# Characteristic Neural Ordinary Differential Equations

## Abstract

We propose Characteristic-Neural Ordinary Differential Equations (C-NODEs), a
framework for extending Neural Ordinary Differential Equations (NODEs) beyond
ODEs. While NODEs model the evolution of latent variables as the solution to an
ODE, C-NODE models the evolution of the latent variables as the solution of a
family of first-order quasi-linear partial differential equations (PDEs) along curves
on which the PDEs reduce to ODEs, referred to as characteristic curves. This in
turn allows the application of the standard frameworks for solving ODEs, namely
the adjoint method. Learning optimal characteristic curves for given tasks improves
the performance and computational efficiency, compared to state of the art NODE
models. We prove that the C-NODE framework extends the classical NODE on
classification tasks by demonstrating explicit C-NODE representable functions
not expressible by NODEs. Additionally, we present C-NODE-based continuous
normalizing flows, which describe the density evolution of latent variables along
multiple dimensions. Empirical results demonstrate the improvements provided
by the proposed method for classification and density estimation on CIFAR-10,
SVHN, and MNIST datasets under a similar computational budget as the existing
NODE methods. The results also provide empirical evidence that the learned
curves improve the efficiency of the system through a lower number of parameters
and function evaluations compared with baselines.

## 1 Introduction

Deep learning and differential equations share many connections, and techniques in the intersection
have led to insights in both fields. One predominant connection is based on certain neural network
architectures resembling numerical integration schemes, leading to the development of Neural
Ordinary Differential Equations (NODEs) [5]. NODEs use a neural network parameterization of
an ODE to learn a mapping from observed variables to a latent variable that is the solution to the
learned ODE. A central benefit of NODEs is the constant memory cost, where backward passes are
computed using the adjoint sensitivity method rather than backpropagating through individual forward
solver steps. Backpropagating through adaptive differential equation solvers to train large NODEs
will often result in memory outage, as mentioned in [5]. Moreover, NODEs provide a flexible
probability density representation often referred to as *continuous normalizing flows* (CNFs). However,
since NODEs can only represent solutions to ODEs, the class of functions is somewhat limited
and may not apply to more general problems that do not have smooth and one-to-one mappings.
To address this limitation, a series of analyses based on methods from differential equations have
been employed to enhance the representation capabilities of NODEs, such as the technique of
controlled differential equations [24], learning higher-order ODEs [32], augmenting dynamics [10],
and considering dynamics with delay terms [55]. Moreover, certain works consider generalizing the
ODE case to partial differential equations (PDEs), such as in [40, 44]. However, these methods do
not use the adjoint method, removing the primary advantage of constant memory cost. This leads us
to the central question motivating the work: can we combine the benefits of the rich function class of

PDEs with the efficiency of the adjoint method? To do so, we propose a method of continuous-depth neural networks that solves a PDE over parametric curves that reduce the PDE to an ODE. Such curves are known as *characteristics*, and they define the solution of the PDE in terms of an ODE [15]. The proposed Characteristic Neural Ordinary Differential Equations (C-NODE) learn both the characteristics and the ODE along the characteristics to solve the PDE over the data space. This allows for a richer class of models while still incorporating the same memory efficiency of the adjoint method. The proposed C-NODE is also an extension of existing methods, as it improves the empirical accuracy of these methods in classification tasks and image quality in generation tasks.

## 2  Related Work

We discuss the related work from both machine learning and numerical analysis perspectives.

### 2.1  Machine Learning and ODEs

NODE is often motivated as a continuous form of a Residual Network (ResNet) [17], since the ResNet can be seen as a forward Euler integration scheme on the latent state [48]. Specifically, a ResNet is composed of multiple blocks where each block can be represented as:

$$u_{t+1} = u_t + f(u_t, \theta),$$

where $u_t$ is the evolving hidden state at time $t$ and $f(u_t, \theta)$ represents the gradient at time $t$, namely $\frac{du}{dt}(u_t)$. Generalizing the model to a step size given by $\Delta t$, we have:

$$u_{t+\Delta t} = u_t + f(u_t, \theta)\Delta t.$$

To adapt this model to a continuous setting, we let $\Delta t \to 0$ and obtain:

$$\lim_{\Delta t \to 0} \frac{u_{t+\Delta t} - u_t}{\Delta t} = \frac{du(t)}{dt}.$$

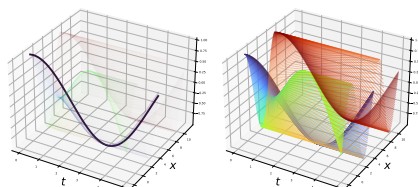

Single Characteristic (ODE)   Multiple Characteristics (PDE)

Figure 1: Comparison of traditional NODE (left) and proposed C-NODE (right). The solution to NODE is the solution to a single ODE, whereas C-NODE represents a series of ODEs that form the solution to a PDE. Each color in C-NODE represents the solution to an ODE with a different initial condition. NODE represents a single ODE, and can only represent $u(x, t)$ along one dimension, for example, $u(x = 0, t)$.

The model can then be evaluated through existing numerical integration techniques, as proposed by [5]:

$$u(t_1) = u(t_0) + \int_{t_0}^{t_1} \frac{du(t)}{dt}(u(t), t)\mathrm{d}t = u(t_0) + \int_{t_0}^{t_1} f(u(t), t, \theta)\mathrm{d}t.$$

Numerical integration can then be treated as a black box, using numerical schemes beyond the forward Euler to achieve higher numerical precision. However, since black box integrators can take an arbitrary number of intermediate steps, backpropagating through individual steps would require too much memory since the individual steps must be saved. Chen et al. [5] addressed this problem by using adjoint backpropagation, which has a constant memory usage. For a given loss function on the terminal state of the hidden state $\mathcal{L}(u(t_1))$, the adjoint $a(t)$ is governed by another ODE:

$$\frac{da(t)}{dt} = -a(t)^{\intercal} \frac{\partial f(u(t), t, \theta)}{\partial u}, \quad a(t_1) = \frac{\partial \mathcal{L}}{\partial u(t_1)},$$

that dictates the gradient with respect to the parameters. The loss $\mathcal{L}(u(t_1))$ can then be calculated by solving another ODE (the adjoint) rather than backpropagating through the calculations involved in the numerical integration.

However, the hidden state governed by an ODE imposes a limitation on the expressiveness of the mapping. For example, Dupont et al. [10] describes a notable limitation of NODEs is in the inability to represent dynamical systems with intersecting trajectories. In response to such limitations, many works have tried to increase the expressiveness of the mapping. Dupont et al. [10] proposed to solve the intersection trajectories problem by augmenting the vector space, lifting the points into additional dimensions; Zhu et al. [55] included time delay in the equation to represent dynamical systems of

greater complexity; Massaroli et al. [32] proposed to condition the vector field on the inputs, allowing the integration limits to be conditioned on the input; Massaroli et al. [32] and Norcliffe et al. [35] additionally proposed and proved a second-order ODE system can efficiently solve the intersecting trajectories problem.

Multiple works have attempted to expand NODE systems to other common differential equation formulations. Sun et al. [44] employed a dictionary method and expanded NODEs to a PDE case, achieving high accuracies both in approximating PDEs and in classifying real-world image datasets. However, Sun et al. [44] suggested that the method is unstable when training with the adjoint method and therefore is unable to make use of the benefits that come with training with adjoint. Zhang et al. [53] proposed a normalizing flow approach based on the Monge-Ampere equation. However, Zhang et al. [53] did not consider using adjoint-based training. Long et al. [30, 31], Raissi et al. [37], Brunton et al. [3] considered discovering underlying hidden PDEs from data and predict dynamics of complex systems. Kidger et al. [24], Morrill et al. [33, 34] used ideas from rough path theory and controlled differential equations to propose a NODE architecture as a continuous recurrent neural network framework. Multiple works have expanded to the stochastic differential equations setting and developed efficient optimization methods for them [16, 22, 23, 25, 26, 28, 29, 49]. Salvi et al. [41] considered stochastic PDEs for spatio-temporal dynamics prediction. Additionally, Chen et al. [6] models spatio-temporal data using NODEs, and Rubanova et al. [39], De Brouwer et al. [8] makes predictions on time series data using NODEs. Physical modeling is also a popular application of NODEs, as control problems are often governed by latent differential equations that can be discovered with data driven methods [7, 14, 51, 54].

NODE systems have also been used for modeling the flow from a simple probability density to a complicated one [5]. Specifically, if $u(t) \in \mathbb{R}^n$ follows the ODE $du(t)/dt = f(u(t))$, where $f(u(t)) \in \mathbb{R}^n$, then its log likelihood from [5, Appendix A] is given by:

$$\frac{\partial \log p(u(t))}{\partial t} = -\operatorname{tr}\left(\frac{df}{du(t)}\right). \tag{1}$$

The trace can be calculated efficiently with a Hutchinson trace estimator [13]. Subsequent work uses invertible ResNet, optimal transport theory, among other techniques to further improve the performance of CNFs [1, 2, 4, 11, 18, 20, 21, 46, 50, 53]. CNF is desirable for having no constraints on the type of neural network used, unlike discrete normalizing flows, which often have constraints on the structure of the latent features [9, 36, 38]. CNFs also inspire development in other generative modeling methods. For instance, a score-based generative model can be seen as a probability flow modeled with an ODE [42, 47].

## 3  Method

We describe the proposed C-NODE method in this section by first providing a brief introduction to the method of characteristics (MoC) for solving PDEs with an illustrative example. We then discuss how we apply the MoC to our C-NODE framework. We finally discuss the types of PDEs we can describe using this method.

### 3.1  Method of Characteristics

The MoC provides a procedure for transforming certain PDEs into ODEs along paths known as *characteristics*. In the most general sense, the method applies to general hyperbolic differential equations; however, for illustration purposes, we will consider a canonical example using the inviscid Burgers equation. A complete exposition on the topic can be found in [15, Chapter 9], but we introduce some basic concepts here for completeness. Let $u(x,t) : \mathbb{R} \times \mathbb{R}_+ \to \mathbb{R}$ satisfy the following inviscid Burgers equation

$$\frac{\partial u}{\partial t} + u\frac{\partial u}{\partial x} = 0, \tag{2}$$

where we dropped the dependence on $x$ and $t$ for ease of notation. We are interested in the solution of $u$ over some bounded domain $\Omega \subset \mathbb{R} \times \mathbb{R}_+$. Consider parametric forms for the spatial component $x(s) : [0, T] \to \mathbb{R}$ and temporal components $t(s) : [0, T] \to \mathbb{R}_+$ over the fictitious variable $s \in [0, T]$.

Intuitively, this allows us to solve an equation on curves $x, t$ as functions of a variable $s$ which we denote $(x(s), t(s))$ as the *characteristic*. Expanding, and writing d as the total derivative, we get

$$\frac{\mathrm{d}}{\mathrm{d}s} u(x(s), t(s)) = \frac{\partial u}{\partial x}\frac{dx}{ds} + \frac{\partial u}{\partial t}\frac{dt}{ds}. \tag{3}$$

Recalling the original PDE in (2) and substituting the proper terms into (3) for $dx/ds = u$, $dt/ds = 1$, $\mathrm{d}u/\mathrm{d}s = 0$, we then recover (2). Note that we now have a system of 3 ODEs, which we can solve to obtain the characteristics as $x(s) = us + x_0$ and $t(s) = s + t_0$ as functions of initial conditions $x_0, t_0$. Finally, by solving over a grid of initial conditions $\{x_0^{(i)}\}_{i=1}^{\infty} \in \partial\Omega$, we can obtain the solution of the PDE over $\Omega$. Putting it all together, we have a new ODE that is written as

$$\frac{\mathrm{d}}{\mathrm{d}s} u(x(s), t(s)) = \frac{\partial u}{\partial t} + u\frac{\partial u}{\partial x} = 0,$$

where we can integrate over $s$ through

$$u(x(T), t(T); x_0, t_0) := \int_0^T \frac{\mathrm{d}}{\mathrm{d}s} u(x(s), t(s))\mathrm{d}s$$

$$:= \int_0^T \frac{\mathrm{d}}{\mathrm{d}s} u(us + x_0, s)\mathrm{d}s,$$

using the adjoint method with boundary conditions $x_0, t_0$. This contrasts the usual direct integration over the variable $t$ that is done in NODE; we now jointly couple the integration through the characteristics. An example of solving this equation over multiple initial conditions is given in Figure 1 with the contrast to standard NODE integration.

To provide some intuition for using MoC, we note that MoC most generally applies to hyperbolic PDEs. The transport equation is an example of this family of PDEs, which roughly describes the propagation of physical quantities through time. Such equations are appropriate for deep learning tasks due to their ability to transport data into different regions of the state space. For instance, in a classification task, we consider the problem of transporting high-dimensional data points that are not linearly separable to spaces where they are linearly separable. Similarly, in generative modeling, we transport a base distribution to data distribution.

### 3.2 Neural Representation of Characteristics

In the proposed method, we learn the components involved in the MoC, namely the characteristics and the function coefficients. We now generalize the example given in 3.1, which involved two variables, to a $k$-dimensional system. Specifically, consider the following nonhomogeneous boundary value problem (BVP)

$$\begin{cases} \frac{\partial \mathbf{u}}{\partial t} + \sum_{i=1}^k a_i(x_1, ..., x_k, \mathbf{u})\frac{\partial \mathbf{u}}{\partial x_i} = \mathbf{c}(x_1, ..., x_k, \mathbf{u}), & \text{on } \mathbf{x}, t \in \mathbb{R}^k \times [0, \infty) \\ \mathbf{u}(\mathbf{x}(0)) = \mathbf{u}_0, & \text{on } \mathbf{x} \in \mathbb{R}^k. \end{cases} \tag{4}$$

Here, $\mathbf{u} : \mathbb{R}^k \to \mathbb{R}^n$ is a multivariate map, $a_i : \mathbb{R}^{k+n} \to \mathbb{R}$ and $\mathbf{c} : \mathbb{R}^{k+n} \to \mathbb{R}^n$ be functions dependent on values of $\mathbf{u}$ and $x$'s. This problem is well-defined and has a solution so long as $\sum_{i=1}^k a_i \frac{\partial \mathbf{u}}{\partial x_i}$ is continuous [12].

MoC has historically been used in a scalar context, but generalization to the vector case is relatively straightforward. A proof of the generalization can be found in Appendix B.1. We decompose the PDE in (4) into the following system of ODEs

$$\frac{dx_i}{ds} = a_i(x_1, ..., x_k, \mathbf{u}), \tag{5}$$

$$\frac{d\mathbf{u}}{ds} = \sum_{i=1}^k \frac{\partial \mathbf{u}}{\partial x_i}\frac{dx_i}{ds} = \mathbf{c}(x_1, ..., x_k, \mathbf{u}). \tag{6}$$

We represent this ODE system by parameterizing $dx_i/ds$ and $\partial \mathbf{u}/\partial x_i$ with neural networks. Consequently, $d\mathbf{u}/ds$ is evolving according to (6).

Following this expansion, we arrive at

$$\mathbf{u}(\mathbf{x}(T)) = \mathbf{u}(\mathbf{x}(0)) + \int_0^T \frac{\mathrm{d}\mathbf{u}}{\mathrm{d}s}(\mathbf{x}, \mathbf{u})\,\mathrm{d}s \tag{7}$$

$$= \mathbf{u}(\mathbf{x}(0)) + \int_0^T [\mathbf{J}_\mathbf{x}\mathbf{u}](\mathbf{x}, \mathbf{u}; \Theta_2)\frac{d\mathbf{x}}{ds}(\mathbf{x}, \mathbf{u}; \Theta_2)\,\mathrm{d}s,$$

where we remove $\mathbf{u}$'s dependency on $\mathbf{x}(s)$ and $\mathbf{x}$'s dependency on $s$ for simplicity of notation. In Equation (7), the functions $\mathbf{J}_\mathbf{x}\mathbf{u}$ and $d\mathbf{x}/ds$ are learnable functions which are the outputs of deep neural networks with inputs $\mathbf{x}$, $\mathbf{u}$ and parameters $\Theta_2$.

### 3.3  Conditioning on data

Previous works primarily modeled the task of classifying a set of data points with a fixed differential equation, neglecting possible structural variations lying in the data. Here, we condition C-NODE on each data point, thereby solving a PDE with a different initial condition. Specifically, consider the term given by the integrand in (7). The neural network representing the characteristic $d\mathbf{x}/ds$ is conditioned on the input data $\mathbf{z} \in \mathbb{R}^w$. Define a feature extractor function $\mathbf{g}(\cdot) : \mathbb{R}^w \to \mathbb{R}^n$ and we have

$$\frac{dx_i}{ds} = a_i(x_1, \ldots, x_k, \mathbf{u}; \mathbf{g}(\mathbf{z})). \tag{8}$$

By introducing $\mathbf{g}(\mathbf{z})$ in (8), the equation describing the characteristics changes depending on the current data point. This leads to the classification task being modeled with a family rather than one single differential equation.

### 3.4  Training C-NODEs

After introducing the main components of C-NODEs, we can integrate them into a unified algorithm. To motivate this section, and to be consistent with part of the empirical evaluation, we will consider classification tasks with data $\{(\mathbf{z}_j, \mathbf{y}_j)\}_{j=1}^N$, $\mathbf{z}_j \in \mathbb{R}^w$, $\mathbf{y}_j \in \mathbb{Z}^+$. For instance, $\mathbf{z}_j$ may be an image, and $\mathbf{y}_j$ is its class label. In the approach we pursue here, the image $\mathbf{z}_j$ is first passed through a feature extractor function $\mathbf{g}(\cdot; \Theta_1) : \mathbb{R}^w \to \mathbb{R}^n$ with parameters $\Theta_1$. The output of $\mathbf{g}$ is the feature $\mathbf{u}_0^{(j)} = \mathbf{g}(\mathbf{z}_j; \Theta_1)$ that provides the boundary condition for the PDE on $\mathbf{u}^{(j)}$. We integrate along different characteristic curves indexed by $s \in [0, T]$ with boundary condition $\mathbf{u}^{(j)}(\mathbf{x}(0)) = \mathbf{u}_0^{(j)}$, and compute the end values as given by (7), where we mentioned in Section 3.2,

$$\mathbf{u}^{(j)}(\mathbf{x}(T)) = \mathbf{u}_0^{(j)} + \int_0^T \mathbf{J}_\mathbf{x}\mathbf{u}^{(i)}\left(\mathbf{x}, \mathbf{u}^{(j)}; \Theta_2\right)\frac{d\mathbf{x}}{ds}\left(\mathbf{x}, \mathbf{u}^{(j)}; \mathbf{u}_0^{(j)}; \Theta_2\right)\mathrm{d}s \tag{9}$$

Finally, $\mathbf{u}^{(j)}(\mathbf{x}(T))$ is passed through another neural network, $\Phi(\mathbf{u}^{(j)}(\mathbf{x}(T)); \Theta_3)$ with input $\mathbf{u}^{(j)}(\mathbf{x}(T))$ and parameters $\Theta_3$ whose output are the probabilities of each class labels for image $\mathbf{z}_j$. The entire learning is now is reduced to finding optimal weights $(\Theta_1, \Theta_2, \Theta_3)$ which can be achieved by minimizing the loss

$$\mathcal{L} = \sum_{j=1}^N L(\Phi(\mathbf{u}^{(j)}(\mathbf{x}(T)); \Theta_3), \mathbf{y}_j),$$

where $L(\cdot)$ is a loss function of choice. In Algorithm 1, we illustrate the implementation procedure with the forward Euler method for simplicity for the framework but note any ODE solver can be used.

### 3.5  Combining MoC with Existing NODE Modifications

As mentioned in the Section 2, the proposed C-NODEs method can be used as an extension to existing NODE frameworks. In all NODE modifications, the underlying expression of $\int_a^b \mathbf{f}(t, \mathbf{u}; \Theta)dt$ remains the same. Modifying this expression to $\int_a^b \mathbf{J}_\mathbf{x}\mathbf{u}(\mathbf{x}, \mathbf{u}; \Theta)d\mathbf{x}/ds(\mathbf{x}, \mathbf{u}; \mathbf{u}_0; \Theta)ds$ results in the proposed C-NODE architecture, with the size of $\mathbf{x}$ being a hyperparameter.

---
**Algorithm 1** C-NODE algorithm using the forward Euler method
---
  **for** each input data $\mathbf{z}_j$ **do**
      extract image feature $\mathbf{u}(s = 0) = \mathbf{g}(\mathbf{z}_j; \Theta_1)$ with a feature extractor neural network.
      **procedure** Integration along $s = 0 \to 1$
      **for** each time step $s_m$ **do**
          calculate $\frac{d\mathbf{x}}{ds}(\mathbf{x}, \mathbf{u}; \mathbf{g}(\mathbf{z}_j; \Theta_1); \Theta_2)$ and $\mathbf{J_x u}(\mathbf{x}, \mathbf{u}; \Theta_2)$.
          calculate $\frac{d\mathbf{u}}{ds} = \mathbf{J_x u} \frac{d\mathbf{x}}{ds}$.
          calculate $\mathbf{u}(s_{m+1}) = \mathbf{u}(s_m) + \frac{d\mathbf{u}}{ds}(s_{m+1} - s_m)$.
      **end for**
      **end procedure**
      classify $\mathbf{u}(s = 1)$ with neural network $\Phi(\mathbf{u}(\mathbf{x}(s = 1)), \Theta_3)$.
  **end for**
---

## 4 Properties of C-NODEs

C-NODE has a number of theoretical properties that contribute to its expressiveness. We provide some theoretical results on these properties in the proceeding sections. We also define continuous normalizing flows (CNFs) with C-NODEs, extending the CNFs originally defined with NODEs.

### 4.1 Intersecting trajectories

As mentioned in [10], one limitation of NODE is that the mappings cannot represent intersecting dynamics. We prove by construction that the C-NODEs can represent some dynamical systems with intersecting trajectories in the following proposition:

**Proposition 4.1.** *The C-NODE can represent a dynamical system on $u(s)$, $du/ds = \mathcal{G}(s, u)$ : $\mathbb{R}_+ \times \mathbb{R} \to \mathbb{R}$, where when $u(0) = 1$, then $u(1) = u(0) + \int_0^1 \mathcal{G}(s, u)ds = 0$; and when $u(0) = 0$, then $u(1) = u(0) + \int_0^1 \mathcal{G}(s, u)ds = 1$.*

*Proof.* See Appendix B.2. ∎

### 4.2 Density estimation with C-NODEs

C-NODEs can also be used to define a continuous density flow that models the density of a variable over space subject to the variable satisfying a PDE. Similar to the change of log probability of NODEs, as in (1), we provide the following proposition for C-NODEs:

**Proposition 4.2.** *Let $u(s)$ be a finite continuous random variable with probability density function $p(u(s))$ and let $u(s)$ satisfy $\frac{du(s)}{ds} = \sum_{i=1}^k \frac{\partial u}{\partial x_i} \frac{dx_i}{ds}$. Assuming $\frac{\partial u}{\partial x_i}$ and $\frac{dx_i}{ds}$ are uniformly Lipschitz continuous in $u$ and continuous in $s$, then the evolution of the log probability of $u$ follows:*

$$\frac{\partial \log p(u(s))}{\partial s} = -\mathrm{tr}\left( \frac{\partial}{\partial u} \sum_{i=1}^k \frac{\partial u}{\partial x_i} \frac{dx_i}{ds} \right)$$

*Proof.* See Appendix B.3. ∎

CNFs are continuous and invertible one-to-one mappings onto themselves, i.e., homeomorphisms. Zhang et al. [52] proved that vanilla NODEs are not universal estimators of homeomorphisms, and augmented neural ODEs (ANODEs) are universal estimators of homeomorphisms. We demonstrate that C-NODEs are pointwise estimators of homeomorphisms, which we formalize in the following proposition:

**Proposition 4.3.** *Given any homeomorphism $h : \Upsilon \to \Upsilon$, $\Upsilon \subset \mathbb{R}^p$, initial condition $u_0$, and time $T > 0$, there exists a flow $u(s, u_0) \in \mathbb{R}^n$ following $\frac{du}{ds} = \frac{\partial u}{\partial x} \frac{dx}{ds} + \frac{\partial u}{\partial t} \frac{dt}{ds}$ such that $u(T, u_0) = h(u_0)$.*

*Proof.* See Appendix B.4. ∎

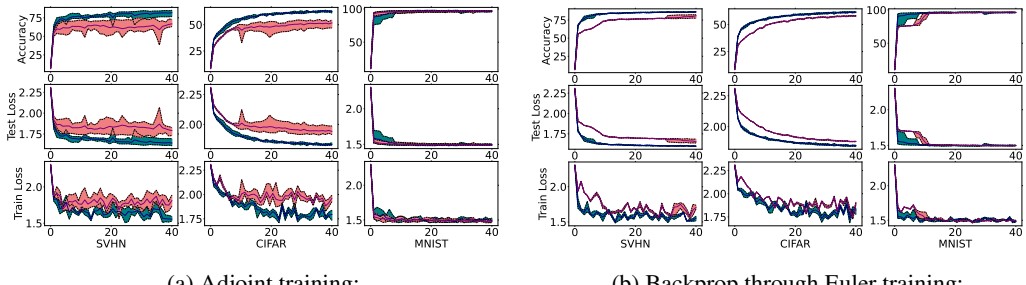

(a) Adjoint training;  (b) Backprop through Euler training;

Figure 2: **Red: NODE. Blue: C-NODE.** Training dynamics of different datasets with adjoint in Fig. 2a and with Euler in Fig. 2b averaged over five runs. The first column is the training process of SVHN, the second column is of CIFAR-10, and the third column is of MNIST. By incorporating the C-NODE method, we achieve a more stable training process in both CIFAR-10 and SVHN, while achieving higher accuracy. Full-sized figure in supplementary materials.

## 5 Experiments

We present experiments on image classification tasks on benchmark datasets, image generation tasks on benchmark datasets, PDE modeling, and time series prediction.

### 5.1 Classification Experiments with Image Datasets

We first conduct experiments for classification tasks on high-dimensional image datasets, including MNIST, CIFAR-10, and SVHN. We provide results for C-NODE and also combine the framework with existing methods, including ANODEs [10], Input Layer NODEs (IL-NODEs) [32], and 2nd-Order NODEs [32]. For all classification experiments, we set the encoder of input images for conditioning to be identity, i.e., $g(z) = z$, making the input into C-NODE the original image. This way, we focus exclusively on the performance of C-NODE.

The results for the experiments with the adjoint method are reported in Table 1 and in Figure 2a. We investigate the performances of the models on classification accuracy and the number of function evaluations (NFE) taken in the adaptive numerical integration. NFE is an indicator of the model's computational complexity, and can also be interpreted as the network depth for the continuous NODE system [5]. Using a similar number of parameters, combining C-NODEs with different models consistently results in higher accuracies and mostly uses smaller numbers of NFEs, indicating a better parameter efficiency. An ablation study on C-NODEs' and NODEs' parameters can be found in Appendix C.2. The performance improvements can be observed, especially on CIFAR-10 and SVHN, where it seems the dynamics to be learned are too complex for ODE systems, requiring a sophisticated model and a large number of NFEs. It appears that solving a PDE system along a multidimensional characteristic is beneficial for training more expressive functions with less complex dynamics, as can be seen in Figures 2a, 2b.

We also report training results using a traditional backpropagation through the forward Euler solver in Figure 2b. The experiments are performed using the same network architectures as the previous experiments using the adjoint method. It appears that C-NODEs converge significantly faster than the NODEs (usually in one epoch) and generally have a more stable training process with smaller variance. In experiments with MNIST, C-NODEs converge in only one epoch, while NODEs converge in roughly 15 epochs. This provides additional empirical evidence on the benefits of training using the characteristics. As shown in Figures 2a, 2b, compared to training with the adjoint method, training with the forward Euler solver results in less variance, indicating a more stable training process. At the same time, training with the adjoint method results in more accurate models, as the adjoint method uses a constant amount of memory, and can employ more accurate adaptive ODE solvers.

### 5.2 Continuous normalizing flow with C-NODEs

We compare the performance of CNFs defined with NODEs to with C-NODEs on MNIST, SVHN, and CIFAR-10. We use a Hutchinson trace estimator to calculate the trace and use multi-scale

| Dataset | Method | Accuracy ↑ | NFE ↓ | Param.[K] ↓ |
|---|---|---|---|---|
| SVHN | NODE | $75.28 \pm 0.836\%$ | 131 | 115.444 |
| | C-NODE | $\mathbf{82.19 \pm 0.478}\%$ | **124** | 113.851 |
| | ANODE | $89.8 \pm 0.952\%$ | 167 | 112.234 |
| | ANODE+C-NODE | $\mathbf{92.23 \pm 0.176}\%$ | **146** | 112.276 |
| | 2nd-Ord | $88.22 \pm 1.11\%$ | 161 | 112.801 |
| | 2nd-Ord+C-NODE | $\mathbf{92.37 \pm 0.118}\%$ | **135** | 112.843 |
| | IL-NODE | $89.69 \pm 0.369\%$ | 195 | 113.368 |
| | IL-NODE+C-NODE | $\mathbf{93.31 \pm 0.088}\%$ | **95** | 113.752 |
| CIFAR-10 | NODE | $56.30 \pm 0.742\%$ | 152 | 115.444 |
| | C-NODE | $\mathbf{64.28 \pm 0.243}\%$ | **151** | 113.851 |
| | ANODE | $70.99 \pm 0.483\%$ | **177** | 112.234 |
| | ANODE+C-NODE | $\mathbf{71.36 \pm 0.220}\%$ | 224 | 112.276 |
| | 2nd-Ord | $70.84 \pm 0.360\%$ | 189 | 112.801 |
| | 2nd-Ord+C-NODE | $\mathbf{73.68 \pm 0.153}\%$ | **131** | 112.843 |
| | IL-NODE | $72.55 \pm 0.238\%$ | 134 | 113.368 |
| | IL-NODE+C-NODE | $\mathbf{73.78 \pm 0.154}\%$ | **85** | 113.752 |
| MNIST | NODE | $96.90 \pm 0.154\%$ | 72 | 85.468 |
| | C-NODE | $\mathbf{97.56 \pm 0.431}\%$ | 72 | 83.041 |
| | ANODE | $99.12 \pm 0.021\%$ | 68 | 89.408 |
| | ANODE+C-NODE | $\mathbf{99.20 \pm 0.002}\%$ | **60** | 88.321 |
| | 2nd-Ord | $99.35 \pm 0.002\%$ | **52** | 89.552 |
| | 2nd-Ord+C-NODE | $\mathbf{99.38 \pm 0.037}\%$ | 61 | 88.465 |
| | IL-NODE | $99.33 \pm 0.039\%$ | **53** | 89.597 |
| | IL-NODE+C-NODE | $\mathbf{99.33 \pm 0.001}\%$ | 60 | 88.51 |

Table 1: Mean test results over 5 runs of different NODE models over SVHN, CIFAR-10, and MNIST. Accuracy and NFE at convergence are reported. Applying C-NODE always increases models' accuracy and usually reduces models' NFE as well as the standard error.

convolutional architectures as done in [9, 13] [1]. Differential equations are solved using the Runge-Kutta method of order 5 of the Dormand-Prince-Shampine solver and trained with the adjoint method. Although the Euler forward method is faster, experimental results show that its fixed step size often leads to negative Bits/Dim, indicating the importance of adaptive solvers. As shown in table 2 and figure 3, using a similar number of parameters, experimental results show that CNFs defined with C-NODEs perform better than CNFs defined with NODEs in terms of Bits/Dim, as well as having lower variance, and using a lower NFE on all of MNIST, CIFAR-10, and SVHN.

## 5.3 PDE modeling with C-NODEs

We consider a synthetic regression example for a hyperbolic PDE with a known solution. Since NODEs assume that the latent state is only dependent on a scalar (namely time), they cannot model dependencies that vary over multiple spatial variables required by most PDEs. We quantify the differences in the representation capabilities by examining how well each method can represent a linear hyperbolic PDE. We also modify the assumptions used in the classification and density estimation experiments where the boundary conditions were constant as in (4). We approximate the following BVP:

$$\begin{cases} u\frac{\partial u}{\partial x} + \frac{\partial u}{\partial t} = u, \\ u(x,0) = 2t, \quad 1 \le x \le 2. \end{cases} \tag{10}$$

---

[1]This is based on the code that the authors of [13] provided in `https://github.com/rtqichen/ffjord`

| Model | MNIST | | | CIFAR-10 | | | SVHN | | |
| --- | --- | --- | --- | --- | --- | --- | --- | --- | --- |
| | B/D | Param. | NFE | B/D | Param. | NFE | B/D | Param. | NFE |
| Real NVP [9] | 1.05 | N/A | – | 3.49 | N/A | – | – | – | – |
| Glow [27] | 1.06 | N/A | – | 3.35 | 44.0M | – | – | – | – |
| RQ-NSF [11] | – | – | – | 3.38 | 11.8M | – | – | – | – |
| Res. Flow [4] | 0.97 | 16.6M | – | **3.28** | 25.2M | – | – | – | – |
| CP-Flow [21] | 1.02 | 2.9M | – | 3.40 | 1.9M | – | – | – | – |
| NODE | 1.00 | **336.1K** | 1350 | 3.49 | 410.1K | 1847 | 2.15 | 410.1K | 1844 |
| C-NODE | **0.95** | 338.0K | **1323** | 3.44 | **406.0K** | **1538** | **2.12** | **406.0K** | **1352** |

Table 2: Experimental results on generation tasks, with NODE, C-NODE, and other models. B/D indicates Bits/dim. Using a similar amount of parameters, C-NODE outperforms NODE on all three datasets, and have a significantly lower NFE when training for CIFAR-10 and SVHN.

(10) has an analytical solution given by $u(x,t) = \frac{2x \exp(t)}{2 \exp(t)+1}$. We generate a training dataset by randomly sampling 200 points $(x,t)$, $x \in [1,2]$, $t \in [0,1]$, as well as values $u(x,t)$ at those points. We test C-NODE and NODE on 200 points randomly sampled as $(x,t) \in [1,2] \times [0,1]$. For this experiment, C-NODE uses 809 parameters while NODE uses 1185 parameters. C-NODE deviates 8.05% from the test dataset, while NODE deviates 30.52%. Further experimental details can be found in Appendix A.3.

## 5.4 Time series prediction with C-NODEs

Finally, we test C-NODEs and NODEs on the time series prediction problem using the MuJoCo dataset [45]. We follow the experimental settings in [39], where we define an autoregressive model with the encoder being an ODE-RNN model and the decoder being a latent ODE[2]. As shown in Figure 5, C-NODEs achieve lower testing mean squared errors (MSEs). After 100 training epochs, C-NODEs achieve 10.14% lower testing MSEs than NODEs.

## 6 Discussion

We describe an approach for extending NODEs to the case of PDEs by solving a series of ODEs along the characteristics of a PDE. The approach applies

Figure 3: **Red: NODE. Blue: C-NODE.** Training dynamics of CNFs on MNIST dataset with adjoint method. We present Bits/dim of the first 50 training epochs.

to any black-box ODE solver and can be combined with existing NODE-based frameworks. We empirically showcase its efficacy on classification tasks while also demonstrating its success in improving convergence using Euler forward method without the adjoint method. Additionally, C-NODE empirically achieves better performances on density estimation tasks, while being more efficient with the number of parameters and using lower NFEs. C-NODE's efficiency over physical modeling and time series prediction is also highlighted with additional experiments.

**Limitations** There are several limitations to the proposed method. The MoC only applies to hyperbolic PDEs, and we only consider first-order semi-linear PDEs in this paper. This may be a limitation since this is a specific class of PDEs that does not model all data. We also did not enforce any particular structure to prevent characteristics from intersecting, which may result in shock waves and rarefactions. However, we believe that this is unlikely to happen due to the high dimensionality of the ambient space. We additionally note that, compared to ANODE, C-NODE's training is not as stable. This can be improved by coupling C-NODEs with ANODEs or other methods.

---

[2]This is based on the code of Rubanova et al. [39] provide in `https://github.com/YuliaRubanova/latent_ode`

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

# A Experimental Details

## A.1 Experimental details of classification tasks

We report the average performance over five independent training processes, and the models are trained for 100 epochs for all three datasets.

The input for 2nd-Ord, NODE, and C-NODE are the original images. In the IL-NODE, we transform the input to a latent space before the integration by the integral; that is, we raise the $\mathbb{R}^{c \times h \times w}$ dimensional input image into the $\mathbb{R}^{(c+p) \times h \times w}$ dimensional latent feature space[3]. We decode the result after performing the continuous transformations along characteristics curves, back to the $\mathbb{R}^{c \times h \times w}$ dimensional object space. Combining this with the C-NODE can be seen as solving a PDE on the latest features of the images rather than on the images directly. We solve first-order PDEs with three variables in CIFAR-10 and SVHN and solve first-order PDEs with two variables in MNIST. The number of parameters of the models is similar by adjusting the number of features used in the networks. We use similar training hyperparameters as [32].

Unlike ODEs, we take derivatives with respect to different variables in PDEs. For a PDE with $k$ variables, this results in the constraint of the balance equations

$$\frac{\partial^2 u}{\partial x_i x_j} = \frac{\partial^2 u}{\partial x_j x_i}, \; i, \, j \in \{1, 2, ..., k\}, i \neq j.$$

This can be satisfied by defining the $k$-th derivative with a neural network, and integrate $k - 1$ times to get the first order derivatives. Another way of satisfying the balance equation is to drop the dependency on the variables, i.e., $\forall i \in \{1, 2, ..., k\}$,

$$\frac{\partial u}{\partial x_i} = f_i(u; \theta).$$

When we drop the dependency, all higher order derivatives are zero, and the balance equations are satisfied.

All experiments were performed on NVIDIA RTX 3090 GPUs on a cloud cluster.

## A.2 Experimental details of continuous normalizing flows

We report the average performance over four independent training processes. As shown in Figure 4, compared to NODE, using a C-NODE structure improves the stability of training, as well as having a better performance. Specifically, the standard errors for C-NODEs on MNIST, SVHN, and CIFAR-10 are 0.37%, 0.51%, and 0.24% respectively, and for NODEs the standard errors on MNIST, SVHN, and CIFAR-10 are 1.07%, 0.32%, and 0.22% respectively.

The experiments are developed using code adapted from the code that the authors of [13] provided in `https://github.com/rtqichen/ffjord`.

All experiments were performed on NVIDIA RTX 3090 GPUs on a cloud cluster.

## A.3 Experimental details of PDE modeling

We want to solve the initial value problem

$$\begin{cases} u\frac{\partial u}{\partial x} + \frac{\partial u}{\partial t} = u, \\ u(x, 0) = 2x, \quad 1 \leq x \leq 2, \end{cases}$$

where the exact solution is $u(x, t) = \frac{2xe^t}{(2e^t + 1)}$. Our dataset's input are 200 randomly sampled points $(x, t)$, $x \in [1, 2]$, $t \in [0, 1]$, and the dataset's outputare the exact solutions at those points.

For the C-NODE architecture, we define four networks: $NN_1(x, t)$ for $\frac{\partial u}{\partial x}$, $NN_2(x, t)$ for $\frac{\partial u}{\partial t}$, $NN_3(t)$ for the characteristic path $(x(s), t(s))$, $NN_4(x)$ for the initial condition. The result is calculated in four steps:

---

[3]This is based on the code of Massaroli et al. [32] provide in `https://github.com/DiffEqML/torchdyn`

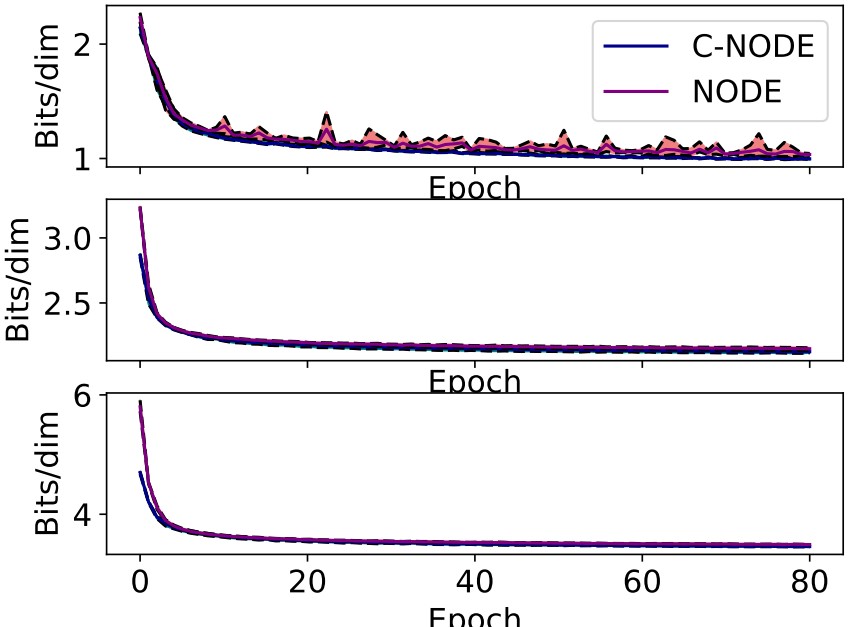

Figure 4: The training process averaged over 4 runs of C-NODE and NODE. The first row are the results on MNIST, the second row are the results on SVHN, the third row are the results on CIFAR-10.

1. Integrate $\Delta u = \int_0^t \frac{du(x(s),t(s))}{ds}ds = \int_0^t \frac{\partial u}{\partial t}\frac{dt}{ds} + \frac{\partial u}{\partial x}\frac{dx}{ds}ds = NN_2 * NN_3[0] + NN_1 * NN_3[1]ds$ as before.

2. Given $x$, $t$, solve equation $\iota + NN_3(NN_4(\iota))[0] * t = x$ for $\iota$ iteratively, with $\iota_{n+1} = x - NN_3(NN_4(\iota_n))[0] * t$. $\iota_0$ is initialized to be $x$.

3. Calculate initial value $u(x(0),t(0)) = NN_4(\iota)$.

4. $u(x,t) = \Delta u + u(x(0),t(0))$.

For the NODE architecture, we define one network: $NN_1(x,t)$ for $\frac{\partial u}{\partial t}$. The result is calculated as $u(x,t) = \int_0^t \frac{\partial u}{\partial t}dt = \int_0^t NN_1 dt$.

All experiments were performed on NVIDIA RTX 3080 ti GPUs on a local machine.

## A.4  Experimental results and details of time series predictions

### A.4.1  Experimental details of time series predictions on MuJoCo dataset

We follow the experimental setup as described in `https://github.com/YuliaRubanova/latent_ode`. NODE's training follows the original setup, with the dimension of the recognition model being 30, the number of units per layer in each of GRU update networks being 100, the number of units per layer in ODE function being 300, the number of layers in ODE function in generative and recognition ODE both being 3.

We use a C-NODE with a dimensionality of 128. The number of units per layer in the network describing $d\mathbf{x}/d\mathbf{s}$ is 12. For the network describing $\partial \mathbf{u}/\partial x_i$, the dimension of the recognition model is 30, the number of units per layer in each of GRU update networks is 100, the number of units per layer in the ODE function is 100, the number of layers in ODE function in generative and recognition ODE is 1.

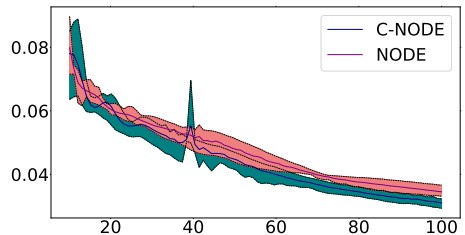

Figure 5: **Red: NODE. Blue: C-NODE.** Training dynamics of ODE-RNNs on the MuJoCo dataset with the "Hopper" model from the Deepmind Control Suit [45]. We present testing mean squared error (MSE) of training epochs 10 to 100. The C-NODE method achieves lower testing MSE while having a lower variance.

### A.4.2 Experiment results of time series predictions on synthetic dataset

We test C-NODEs, ANODEs, and NODEs on a synthetic time series prediction problem. We define a function by $u(x,t) = \frac{2x \exp(t)}{2 \exp(t)+1}$, and we sample $\tilde{u} = u(x,t) + 0.1\epsilon_t$, where $\epsilon_t \sim \mathcal{N}(0,1)$ over $x \in [1,2]$, $t \in [0,1]$ to generate the training dataset. We test the performance on $t \in [n, n+1]$ with $n \in \{0, 1, \ldots, 5\}$. To make the problem more challenging, $x$ values are omitted, and only $t$ values are provided during both training and testing. As shown in Table 3, C-NODE produces more profound improvements over NODEs as time increases.

| Time | [0,1] | [1,2] | [2,3] | [3,4] | [4,5] | [5,6] |
|---|---|---|---|---|---|---|
| NODE | 0.0322 | 0.1764 | 0.4681 | 0.8093 | 1.1911 | 1.6202 |
| ANODE | 0.0428 | 0.0629 | 0.1248 | 0.2778 | 0.5360 | 0.9252 |
| C-NODE | **0.0270** | **0.0365** | **0.0582** | **0.1474** | **0.3300** | **0.6054** |

Table 3: Time series prediction results for NODE, ANODE, and C-NODE at different time intervals. Errors are testing mean squared errors. Across all time intervals, C-NODE outperforms NODE and ANODE.

We also test C-NODEs, NODEs, and ANODEs on time series prediction with different levels of noise. Specifically, using the same function as above, we form training and testing dataset with $\epsilon_t \sim \mathcal{N}(0,m)$, $m \in \{0, 1, \ldots, 5\}$. We test the performance on the time period $t \in [0,1]$.

| Noise Level | 0 | 1 | 2 | 3 | 4 | 5 |
|---|---|---|---|---|---|---|
| NODE | 0.0326 | 0.1784 | 0.7886 | 1.9685 | 3.7530 | 6.1553 |
| ANODE | 0.04 | 0.1984 | 0.6035 | 1.0574 | 1.4850 | **2.0593** |
| C-NODE | **0.0267** | **0.1011** | **0.3294** | **0.7148** | **1.2856** | 2.0834 |

Table 4: Time series prediction results for NODE, ANODE, and C-NODE at different noise levels. Errors are testing mean squared errors.

### A.4.3 Experimental details of time series predictions on synthetic dataset

We want to predict $u(x,t) = \frac{2 \cdot x \cdot e^t}{2 \cdot e^t + 1}$ at different time $t$, with $x \in [1,2]$, and $x$ being not accessible to the network. We also provide the network with the value of $u(1,0)$.

We use a 8 dimensional C-NODE network. The result is calculated with

$$u(x,t) = u(1,0) + \int_0^t \sum_{i=1}^{8} \frac{\partial u}{\partial z_i} \frac{dz_i}{ds} ds.$$

513 NODE is calculated with

$$u(x,t) = u(1,0) + \int_0^t \frac{\partial u}{\partial t} dt.$$

514 In our experiments, C-NODEs use 1221 parameters, ANODEs use 1270 parameters, NODEs use
515 1290 parameters.

516 All experiments were performed on NVIDIA RTX 3080 ti GPUs on a local machine.

## B Approximation Capabilities of C-NODE

518 **Proposition B.1** (Method of Characteristics for Vector Valued PDEs). *Let* $\mathbf{u}(x_1, \ldots, x_k) : \mathbb{R}^k \to \mathbb{R}^n$
519 *be the solution of a first order semilinear PDE on a bounded domain* $\Omega \subset \mathbb{R}^k$ *of the form*

$$\sum_{i=1}^k a_i(x_1, \ldots, x_k, \mathbf{u}) \frac{\partial \mathbf{u}}{\partial x_i} = \mathbf{c}(x_1, \ldots, x_k, \mathbf{u}) \quad on \ (x_1, \ldots, x_k) = \mathbf{x} \in \Omega. \tag{11}$$

520 *Additionally, let* $\mathbf{a} = (a_1, \ldots, a_k)^T : \mathbb{R}^{k+n} \to \mathbb{R}^k, \mathbf{c} : \mathbb{R}^{k+n} \to \mathbb{R}^n$ *be Lipschitz continuous*
521 *functions. Define a system of ODEs as*

$$\begin{cases} \frac{d\mathbf{x}}{ds}(s) &= \mathbf{a}(\mathbf{x}(s), \mathbf{U}(s)) \\ \frac{d\mathbf{U}}{ds}(s) &= \mathbf{c}(\mathbf{x}(s), \mathbf{U}(s)) \\ \mathbf{x}(0) &:= \mathbf{x}_0, \ \mathbf{x}_0 \in \partial\Omega \\ \mathbf{u}(\mathbf{x}_0) &:= \mathbf{u}_0 \\ \mathbf{U}(0) &:= \mathbf{u}_0 \end{cases}$$

522 *where* $\mathbf{x}_0$ *and* $\mathbf{u}_0$ *define the initial condition,* $\partial\Omega$ *is the boundary of the domain* $\Omega$*. Given initial*
523 *conditions* $\mathbf{x}_0, \mathbf{u}_0$*, the solution of this system of ODEs* $\mathbf{U}(s) : [a, b] \to \mathbb{R}^d$ *is equal to the solution of*
524 *the PDE in Equation* (11) *along the characteristic curve defined by* $\mathbf{x}(s)$*, i.e.,* $\mathbf{u}(\mathbf{x}(s)) = \mathbf{U}(s)$*. The*
525 *union of solutions* $\mathbf{U}(s)$ *for all* $\mathbf{x}_0 \in \partial\Omega$ *is equal to the solution of the original PDE in Equation*
526 (11) *for all* $\mathbf{x} \in \Omega$*.*

**Lemma B.2** (Gronwall's Lemma [19]). *Let* $U \subset \mathbb{R}^n$ *be an open set. Let* $\mathbf{f} : U \times [0, T] \to \mathbb{R}^n$ *be a*
*continuous function and let* $\mathbf{h_1}, \mathbf{h_2} : [0, T] \to U$ *satisfy the initial value problems:*

$$\frac{d\mathbf{h_1}(t)}{dt} = f(\mathbf{h_1}(t), t), \ \mathbf{h_1}(0) = \mathbf{x_1},$$

$$\frac{d\mathbf{h_2}(t)}{dt} = f(\mathbf{h_2}(t), t), \ \mathbf{h_2}(0) = \mathbf{x_2}.$$

*If there exists non-negative constant* $C$ *such that for all* $t \in [0, T]$

$$\|\mathbf{f}(\mathbf{h_2}(t), t) - \mathbf{f}(\mathbf{h_1}(t), t)\| \le C\|\mathbf{h_2}(t) - \mathbf{h_1}(t)\|,$$

*where* $\| \cdot \|$ *is the Euclidean norm. Then, for all* $t \in [0, T]$*,*

$$\|\mathbf{h_2}(t) - \mathbf{h_1}(t)\| \le e^{Ct}\|\mathbf{x_2} - \mathbf{x_1}\|.$$

### B.1 Proof of Proposition B.1

528 This proof is largely based on the proof for the univarate case provided at[4]. We extend for the vector
529 valued case.

530 *Proof.* For PDE on $\mathbf{u}$ with $k$ input, and an $n$-dimensional output, we have $a_i : \mathbb{R}^{k+n} \to \mathbb{R}, \frac{\partial \mathbf{u}}{\partial x_i} \in \mathbb{R}^n$,
531 and $\mathbf{c} : \mathbb{R}^{k+n} \to \mathbb{R}^n$. In proposition B.1, we look at PDEs in the following form

$$\sum_{i=1}^k a_i(x_1, \ldots, x_k, \mathbf{u}) \frac{\partial \mathbf{u}}{\partial x_i} = \mathbf{c}(x_1, \ldots, x_k, \mathbf{u}). \tag{12}$$

---

[4]https://en.wikipedia.org/wiki/Method_of_characteristics#Proof_for_quasilinear_
Case

532    Defining and substituting $\mathbf{x} = (x_1, \ldots, x_k)^\mathsf{T}$, $\mathbf{a} = (a_1, \ldots, a_k)^\mathsf{T}$, and Jacobian $\mathbf{J}(\mathbf{u}(\mathbf{x})) =$
533    $(\frac{\partial \mathbf{u}}{\partial x_1}, \ldots, \frac{\partial \mathbf{u}}{\partial x_k}) \in \mathbb{R}^{n \times k}$ into Equation (11) result in

$$\mathbf{J}(\mathbf{u}(\mathbf{x}))\mathbf{a}(\mathbf{x}, \mathbf{u}) = \mathbf{c}(\mathbf{x}, \mathbf{u}). \tag{13}$$

From proposition B.1, the characteristic curves are given by

$$\frac{dx_i}{ds} = a_i(x_1, \ldots, x_k, \mathbf{u}),$$

534    and the ODE system is given by

$$\frac{d\mathbf{x}}{ds}(s) = \mathbf{a}(\mathbf{x}(s), \mathbf{U}(s)), \tag{14}$$

535

$$\frac{d\mathbf{U}}{ds}(s) = \mathbf{c}(\mathbf{x}(s), \mathbf{U}(s)). \tag{15}$$

Define the difference between the solution to (15) and the PDE in (11) as

$$\Delta(s) = \|\mathbf{u}(\mathbf{x}(s)) - \mathbf{U}(s)\|^2 = (\mathbf{u}(\mathbf{x}(s)) - \mathbf{U}(s))^\mathsf{T} (\mathbf{u}(\mathbf{x}(s)) - \mathbf{U}(s)),$$

536    Differentiating $\Delta(s)$ with respect to $s$ and plugging in (14), we get

$$\Delta'(s) := \frac{d\Delta(s)}{ds} = 2(\mathbf{u}(\mathbf{x}(s)) - \mathbf{U}(s)) \cdot (\mathbf{J}(\mathbf{u})\mathbf{x}'(s) - \mathbf{U}'(s))$$
$$= 2[\mathbf{u}(\mathbf{x}(s)) - \mathbf{U}(s)] \cdot [\mathbf{J}(\mathbf{u})\mathbf{a}(\mathbf{x}(s), \mathbf{U}(s)) - \mathbf{c}(\mathbf{x}(s), \mathbf{U}(s))]. \tag{16}$$

537    (13) gives us $\sum_{i=1}^{k} a_i(x_1, \ldots, x_k, \mathbf{u})\frac{\partial \mathbf{u}}{\partial x_i} - \mathbf{c}(x_1, \ldots, x_k, \mathbf{u}) = 0$. Plugging this equality into (16)
538    and rearrange terms, we have

$$\Delta'(s) = 2[\mathbf{u}(\mathbf{x}(s)) - \mathbf{U}(s)] \cdot \{[\mathbf{J}(\mathbf{u})\mathbf{a}(\mathbf{x}(s), \mathbf{U}(s)) - \mathbf{c}(\mathbf{x}(s), \mathbf{U}(s))]$$
$$- [\mathbf{J}(\mathbf{u})\mathbf{a}(\mathbf{x}(s), \mathbf{u}(s)) - \mathbf{c}(\mathbf{x}(s), \mathbf{u}(s))]\}.$$

539    Combining terms, we have

$$\Delta' = 2(\mathbf{u} - \mathbf{U}) \cdot ([\mathbf{J}(\mathbf{u})\mathbf{a}(\mathbf{U}) - \mathbf{c}(\mathbf{U})] - [\mathbf{J}(\mathbf{u})\mathbf{a}(\mathbf{u}) - \mathbf{c}(\mathbf{u})])$$
$$= 2(\mathbf{u} - \mathbf{U}) \cdot (\mathbf{J}(\mathbf{u})[\mathbf{a}(\mathbf{U}) - \mathbf{a}(\mathbf{u})] + [\mathbf{c}(\mathbf{U}) - \mathbf{c}(\mathbf{u})]).$$

540    Applying triangle inequality, we have

$$\|\Delta'\| \leq 2\|\mathbf{u} - \mathbf{U}\|(\|\mathbf{J}(\mathbf{u})\|\|\mathbf{a}(\mathbf{U}) - \mathbf{a}(\mathbf{u})\| + \|\mathbf{c}(\mathbf{U}) - \mathbf{c}(\mathbf{u})\|).$$

541    By the assumption in proposition B.1, $\mathbf{a}$ and $\mathbf{c}$ are Lipschitz continuous. By Lipschitz continuity, we
542    have $\|\mathbf{a}(\mathbf{U}) - \mathbf{a}(\mathbf{u}))\| \leq A\|\mathbf{u} - \mathbf{U}\|$ and $\|\mathbf{c}(\mathbf{U}) - \mathbf{c}(\mathbf{u}))\| \leq B\|\mathbf{u} - \mathbf{U}\|$, for some constants A and
543    B in $\mathbb{R}_+$. Also, for compact set $[0, s_0]$, $s_0 < \infty$, since both $\mathbf{u}$ and Jacobian $\mathbf{J}$ are continuous mapping,
544    $\mathbf{J}(\mathbf{u})$ is also compact. Since a subspace of $\mathbb{R}^n$ is compact if and only it is closed and bounded, $\mathbf{J}(\mathbf{u})$
545    is bounded [43]. Thus, $\|\mathbf{J}(\mathbf{u})\| \leq M$ for some constant $M$ in $\mathbb{R}_+$. Define $C = 2(AM + B)$, we
546    have

$$\|\Delta'(s)\| \leq 2(AM\|\mathbf{u} - \mathbf{U}\| + B\|\mathbf{u} - \mathbf{U}\|)\|\mathbf{u} - \mathbf{U}\|$$
$$= C\|\mathbf{u} - \mathbf{U}\|^2$$
$$= C\|\Delta(s)\|.$$

From proposition B.1, we have $\mathbf{u}(\mathbf{x}(0)) = \mathbf{U}(0)$. As proved above, we have

$$\left\| \frac{d\mathbf{u}(\mathbf{x}(s))}{ds} - \frac{d\mathbf{U}(s)}{ds} \right\| := \|\Delta'(s)\| \leq C\|\Delta(s)\|,$$

where $C < \infty$. Thus, by lemma B.2, we have

$$\|\Delta(s)\| \leq e^{Ct}\|\Delta(0)\| = e^{Ct}\|\mathbf{u}(\mathbf{x}(0)) - \mathbf{U}(0)\| = 0.$$

547    This further implies that $\mathbf{U}(s) = \mathbf{u}(\mathbf{x}(s))$, so long as $\mathbf{a}$ and $\mathbf{c}$ are Lipschitz continuous.     $\square$

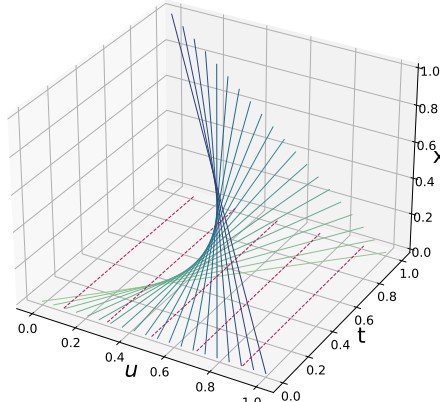

Figure 6: Comparison of C-NODEs and NODEs. C-NODEs (solid blue) learn a family of integration paths conditioned on the input value, avoiding intersecting dynamics. NODEs (dashed red) integrate along a 1D line that is not conditioned on the input value and can not represent functions requiring intersecting dynamics.

## B.2 Proof of Proposition 4.1

*Proof.* Suppose have C-NODE given by

$$\frac{\mathrm{d}u}{\mathrm{d}s} = \frac{\partial u}{\partial x}\frac{dx}{ds} + \frac{\partial u}{\partial t}\frac{dt}{ds}.$$

Write out specific functions for these terms to match the desired properties of the function. Define initial condition $u(0,0) = u_0$. By setting

$$\frac{dx}{ds}(s, u_0, \theta) = 1, \qquad\qquad \frac{dt}{ds}(s, u_0, \theta) = u_0,$$
$$\frac{\partial u}{\partial x}(u(x,t), \theta) = 1, \qquad\qquad \frac{\partial u}{\partial t}(u(x,t), \theta) = -2,$$

have the ODE and solution,

$$\frac{\mathrm{d}u}{\mathrm{d}s} = 1 - 2u_0$$
$$\implies u(s; u_0) = (1 - 2u_0)\,s$$
$$\implies u\left(s; \begin{bmatrix} 0 \\ 1 \end{bmatrix}\right) = \left(1 - 2\begin{bmatrix} 0 \\ 1 \end{bmatrix}\right) s = \begin{bmatrix} 1 \\ -1 \end{bmatrix} s.$$

To be specific, we can represent this system with the following family of PDEs:

$$\frac{\partial u}{\partial x} + u_0 \frac{\partial u}{\partial t} = 1 - 2u_0.$$

We can solve this system to obtain a function that has intersecting trajectories. The solution is visualized in Figure 6, which shows that C-NODE can be used to learn and represent this function $\mathcal{G}$. It should be noted that this is not the only possible solution to function $\mathcal{G}$, as when $\partial t/\partial s = 0$, we fall back to a NODE system with the dynamical system conditioned on the input data. In this conditioned setting, we can then represent $\mathcal{G}$ by stopping the dynamics at different times $t$ as in [32]. $\qquad\square$

## B.3 Proof of Proposition 4.2

The proof uses the change of variables formula for a particle that depends on a vector rather than a scalar and it follows directly from the proof given in [5, Appendix A]. We provide the full proof for completeness.

563 *Proof.* Assume $\sum_{i=1}^{k} \frac{\partial u}{\partial x_i} \frac{dx_i}{ds}$ is Lipschitz continuous in $u$ and continuous in $t$, so every initial value
564 problem has a unique solution [12]. Also assume $u(s)$ is bounded.

565 Want

$$\frac{\partial p(u(s))}{\partial s} = \text{tr}\left(\frac{\partial}{\partial u} \sum_{i=1}^{k} \frac{\partial u}{\partial x_i} \frac{dx_i}{ds}\right).$$

566 Define $T_\epsilon = u(s + \epsilon)$. The discrete change of variables states that $u_1 = f(u_0) \Rightarrow \log p(u_1) =$
567 $\log p(u_0) - \log|\det \frac{\partial f}{\partial u_0}|$ [38].

568 Take the limit of the time difference between $u_0$ and $u_1$, by definition of derivatives,

$$
\begin{aligned}
\frac{\partial \log p(u(s))}{\partial t} &= \lim_{\epsilon \to 0^+} \frac{\log p(u(s+\epsilon)) - \log p(u(s))}{\epsilon} \\
&= \lim_{\epsilon \to 0^+} \frac{\log p(u(s)) - \log|\det \frac{\partial}{\partial u} T_\epsilon(u(t))| - \log p(u(s))}{\epsilon} \\
&= -\lim_{\epsilon \to 0^+} \frac{\log|\det \frac{\partial}{\partial u} T_\epsilon(u(s))|}{\epsilon} \\
&= -\lim_{\epsilon \to 0^+} \frac{\frac{\partial}{\partial \epsilon} \log|\det \frac{\partial}{\partial u} T_\epsilon(u(s))|}{\frac{\partial}{\partial \epsilon} \epsilon} \\
&= -\lim_{\epsilon \to 0^+} \frac{\partial}{\partial \epsilon} \log|\det \frac{\partial}{\partial u} T_\epsilon(u(s))| - \lim_{\epsilon \to 0^+} \frac{\partial}{\partial \epsilon} \log|\det \frac{\partial}{\partial u} T_\epsilon(u(s))| \\
&= -\lim_{\epsilon \to 0^+} \frac{1}{|\det \frac{\partial}{\partial u} T_\epsilon(u(s))|} \frac{\partial}{\partial \epsilon}|\det \frac{\partial}{\partial u} T_\epsilon(u(s))| \\
&= -\frac{\lim_{\epsilon \to 0^+} \frac{\partial}{\partial \epsilon}|\det \frac{\partial}{\partial u} T_\epsilon(u(s))|}{\lim_{\epsilon \to 0^+}|\det \frac{\partial}{\partial u} T_\epsilon(u(s))|} \\
&= -\lim_{\epsilon \to 0^+} \frac{\partial}{\partial \epsilon}|\det \frac{\partial}{\partial u} T_\epsilon(u(s))|
\end{aligned}
$$

569 The Jacobi's formula states that if $A$ is a differentiable map from the real numbers to $n \times n$ matrices,
570 then $\frac{d}{dt} \det A(t) = tr(adj(A(t)) \frac{dA(t)}{dt})$, where $adj$ is the adjugate. Thus, have

$$
\begin{aligned}
\frac{\partial \log p(u(t))}{\partial t} &= -\lim_{\epsilon \to 0^+} \text{tr}\left[adj\left(\frac{\partial}{\partial u} T_\epsilon(u(s))\right) \frac{\partial}{\partial \epsilon} \frac{\partial}{\partial u} T_\epsilon(u(s))\right] \\
&= -\text{tr}\left[\left(\lim_{\epsilon \to 0^+} adj\left(\frac{\partial}{\partial u} T_\epsilon(u(t))\right)\right)\left(\lim_{\epsilon \to 0^+} \frac{\partial}{\partial \epsilon} \frac{\partial}{\partial u} T_\epsilon(u(s))\right)\right] \\
&= -\text{tr}\left[adj\left(\frac{\partial}{\partial u} u(t)\right) \lim_{\epsilon \to 0^+} \frac{\partial}{\partial \epsilon} \frac{\partial}{\partial u} T_\epsilon(u(s))\right] \\
&= -\text{tr}\left[\lim_{\epsilon \to 0^+} \frac{\partial}{\partial \epsilon} \frac{\partial}{\partial u} T_\epsilon(u(s))\right]
\end{aligned}
$$

571  Substituting $T_\epsilon$ with its Taylor series expansion and taking the limit, we have

$$\frac{\partial \log p(u(t))}{\partial t} = -\operatorname{tr}\left(\lim_{\epsilon \to 0^+} \frac{\partial}{\partial \epsilon}\frac{\partial}{\partial u}\left(u + \epsilon\frac{du}{ds} + \mathcal{O}(\epsilon^2) + \mathcal{O}(\epsilon^3) + ...\right)\right)$$

$$= -\operatorname{tr}\left(\lim_{\epsilon \to 0^+} \frac{\partial}{\partial \epsilon}\frac{\partial}{\partial u}\left(u + \epsilon\sum_{i=1}^{k}\frac{\partial u}{\partial x_i}\frac{dx_i}{ds} + \mathcal{O}(\epsilon^2) + \mathcal{O}(\epsilon^3) + ...\right)\right)$$

$$= -\operatorname{tr}\left(\lim_{\epsilon \to 0^+} \frac{\partial}{\partial \epsilon}\left(I + \frac{\partial}{\partial u}\epsilon\sum_{i=1}^{k}\frac{\partial u}{\partial x_i}\frac{dx_i}{ds} + \mathcal{O}(\epsilon^2) + \mathcal{O}(\epsilon^3) + ...\right)\right)$$

$$= -\operatorname{tr}\left(\lim_{\epsilon \to 0^+}\left(\frac{\partial}{\partial u}\sum_{i=1}^{k}\frac{\partial u}{\partial x_i}\frac{dx_i}{ds} + \mathcal{O}(\epsilon) + \mathcal{O}(\epsilon^2) + ...\right)\right)$$

$$= -\operatorname{tr}\left(\frac{\partial}{\partial u}\sum_{i=1}^{k}\frac{\partial u}{\partial x_i}\frac{dx_i}{ds}\right)$$

572  $\square$

### B.4  Proof of Proposition 4.3

574  *Proof.* To prove proposition 4.3, need to show that for any homeomorphism $h(\cdot)$, there exists a
575  $u(s, u_0) \in \mathbb{R}^n$ following a C-NODE system such that $u(s = T, u_0) = h(u_0)$.

576  Without loss of generality, say $T = 1$.

577  Define C-NODE system

$$\begin{cases} \frac{du}{ds} = \frac{\partial u}{\partial x}\frac{dx}{ds} + \frac{\partial u}{\partial t}\frac{dt}{ds}, \\ \frac{dx}{ds}(s, u_0) = 1, \\ \frac{\partial u}{\partial x}(u(x,t)) = h(u_0), \\ \frac{dt}{ds}(s, u_0) = u_0, \\ \frac{\partial u}{\partial t}(u(x,t)) = -1. \end{cases}$$

578  Then, $\frac{du}{ds} = h(u_0) - u_0$. At $s = 1$, have

$$u(s = 1, u_0) = u(s = 0, u_0) + \int_0^1 \frac{du}{ds}ds$$

$$= u_0 + \int_0^1 \frac{\partial u}{\partial x}\frac{dx}{ds} + \frac{\partial u}{\partial t}\frac{dt}{ds}ds$$

$$= u_0 + \int_0^1 h(u_0)\cdot 1 + (-1)\cdot u_0 ds$$

$$= u_0 + h(u_0) - u_0$$

$$= h(u_0).$$

579  The inverse map will be defined by integration backwards. Specifically, have

$$u(s = 0, u_0) = u(s = 1, u_0) + \int_1^0 \frac{du}{ds}ds$$

$$= h(u_0) - \int_0^1 \frac{\partial u}{\partial x}\frac{dx}{ds} + \frac{\partial u}{\partial t}\frac{dt}{ds}ds$$

$$= h(u_0) - \int_0^1 h(u_0)\cdot 1 + (-1)\cdot u_0 ds$$

$$= h(u_0) - h(u_0) + u_0$$

$$= u_0.$$

Thus, for any homeomorphism $h(\cdot)$, there exists a C-NODE system, such that forward integration for time $s = 1$ is equivalent as applying $h(\cdot)$, and backward integration for time $s = 1$ is equivalent to applying $h^{-1}(\cdot)$. □

# C  Ablation Study

## C.1  Ablation study on dimension of C-NODE

We perform an ablation study on the impact of the number of dimensions of the C-NODE we implement. This study allows us to evaluate the relationship between the model performance and the model's limit of mathematical approximating power. Empirical results show that as we increase the number of dimensions used in the C-NODE model, the C-NODE's performance first improves and then declines, due to overfitting. We have found out that information criteria like AIC and BIC can be successfully applied for dimension selection in this scenario.

In previous experiments, we represent $\partial \mathbf{u}/\partial x_i$ with separate and independent neural networks $\mathbf{c}_i(\mathbf{u}, \theta)$. Here, we represent all $k$ functions as a vector-valued function $[\partial \mathbf{u}/\partial x_1, ..., \partial \mathbf{u}/\partial x_k]^T$. We approximate this vector-valued function with a neural network $\mathbf{c}(\mathbf{u}, \theta)$. The model is trained using the Euler solver to have better training stability when the neural network has a large number of parameters. Experiment details for the ablation study is as shown in Figures 7, 8, 9.

## C.2  Ablation study on number of parameters

We show C-NODE's parameter efficiency over NODE with an ablation study on the image classification task on the CIFAR-10 dataset. Specifically, under a similar training setup, we experiment with C-NODE with 95071, 55855, and 17379 parameters and experiment with NODE with 96044, 56828, and 17444 parameters. As shown in Figure 10, although C-NODE has more variance in its performance, it outperforms NODE along the whole training process in all three cases.

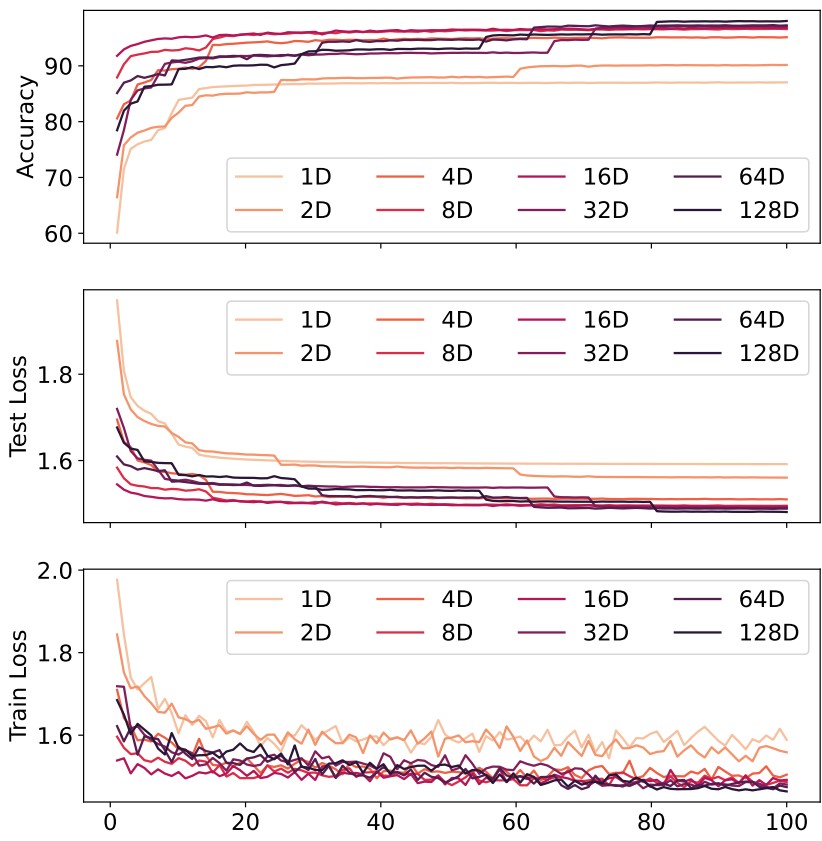

Figure 7: The training process averaged over 4 runs of C-NODE with 1, 2, 4, 8, 16, 32, 64, 128, 256, 512, and 1024 dimensions on the MNIST dataset. The first row is the accuracy of prediction, the second row is the testing error, and the third row is the training error.

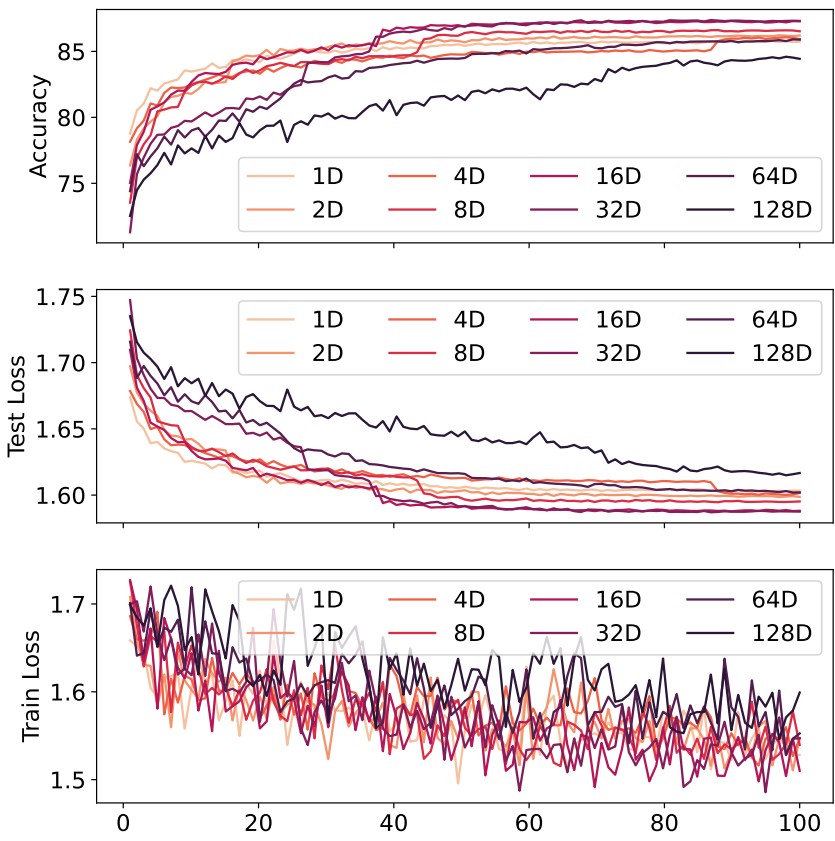

Figure 8: The training process averaged over 4 runs of C-NODE with 1, 2, 4, 8, 16, 32, 64, and 128 dimensions on the SVHN dataset. The first row is the accuracy of prediction, the second row is the testing error, and the third row is the training error.

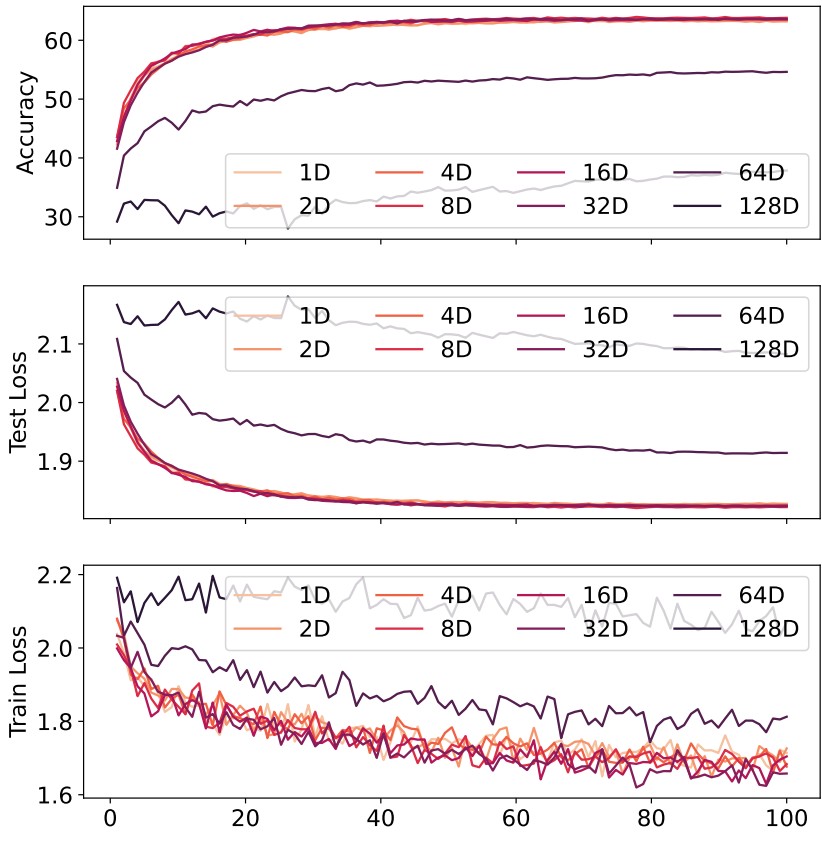

Figure 9: The training process averaged over 4 runs of C-NODE with 1, 2, 4, 8, 16, 32, 64, and 128 dimensions on the CIFAR-10 dataset. The first row is the accuracy of prediction, the second row is the testing error, and the third row is the training error.

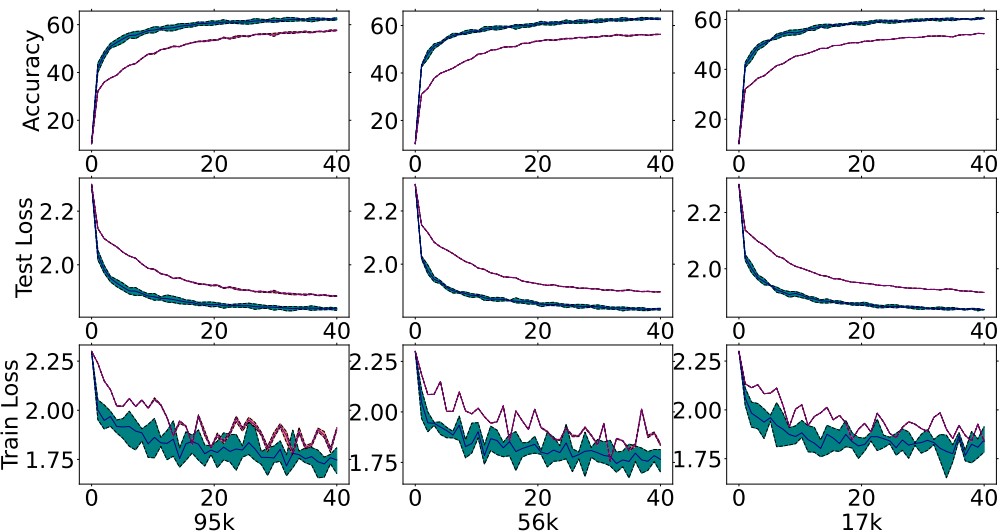

Figure 10: The training process averaged over four runs of C-NODE with 95071, 55855, and 17379 parameters on the CIFAR-10 dataset, and NODE with 96044, 55855, and 17379 parameters. The first row is the prediction accuracy, the second row is the testing error, and the third row is the training error. Blue lines are the results for C-NODE, and red lines are the results for NODE.

# D Algorithm for continuous normalizing flows defined with C-NODE

We additionally provide algorithms for training and sampling CNFs defined with C-NODEs.

---

**Algorithm 2** Algorithm for training CNFs defined with C-NODE

---

given probability density function of $p(\mathbf{u}(s = 0)) = p_0(\cdot)$
**for** each input data $\mathbf{z}_j$ **do**

  Given $\begin{bmatrix} \mathbf{u}(1) \\ \log p(\mathbf{z}_j) - \log p(\mathbf{u}(1)) \end{bmatrix} = \begin{bmatrix} \mathbf{z}_j \\ 0 \end{bmatrix}$

  **procedure** Integrate from $1 \to 0$ to get $\begin{bmatrix} \mathbf{u}(0) \\ \log p(\mathbf{z}_j) - \log p(\mathbf{u}(0)) \end{bmatrix}$

  **for** each time step $s_m$ **do**
    calculate $\frac{d\mathbf{x}}{ds}(\mathbf{x}, \mathbf{u}; \mathbf{g}(\mathbf{z}_j; \Theta_1); \Theta_2)$ and $\mathbf{J_x}\mathbf{u}(\mathbf{x}, \mathbf{u}; \Theta_2)$.
    calculate $\frac{d\mathbf{u}}{ds} = \mathbf{J_x}\mathbf{u}\frac{d\mathbf{x}}{ds}$.
    calculate $-\mathrm{tr}(\frac{\partial}{\partial \mathbf{u}}\mathbf{J_x}\mathbf{u}\frac{d\mathbf{x}}{ds})$ with Hutchinson trace estimator [13].

    calculate $\begin{bmatrix} \mathbf{u}(s_{m+1}) \\ \log p(\mathbf{u}(s_{m+1})) \end{bmatrix} = \begin{bmatrix} \mathbf{u}(s_m) \\ \log p(\mathbf{u}(s_m)) \end{bmatrix} + \begin{bmatrix} \frac{d\mathbf{u}}{dt} \\ \frac{\partial \log p(\mathbf{u}(s))}{\partial s} \end{bmatrix} (s_{m+1} - s_m)$.

  **end for**
  evaluate $p_0(\mathbf{u}(0))$
  calculate $\log p(\mathbf{z}_j) = (\log p(\mathbf{z}_j) - \log p(\mathbf{u}(0))) + \log p_0(\mathbf{u}(0))$
  optimize $\log p(\mathbf{z}_j)$ with an optimization algorithm (stochastic gradient descent etc.)
**end for**

---

---

**Algorithm 3** Algorithm for sampling CNFs defined with C-NODE

---

**procedure** sample $\mathbf{u}(s = 0)$ from base distribution $p_0(\cdot)$
**procedure** Integrate from $0 \to 1$ to get $\mathbf{u}(s = 1)$
**for** each time step $s_m$ **do**
  calculate $\frac{d\mathbf{x}}{ds}(\mathbf{x}, \mathbf{u}; \mathbf{g}(\mathbf{z}_j; \Theta_1); \Theta_2)$ and $\mathbf{J_x}\mathbf{u}(\mathbf{x}, \mathbf{u}; \Theta_2)$.
  calculate $\frac{d\mathbf{u}}{ds} = \mathbf{J_x}\mathbf{u}\frac{d\mathbf{x}}{ds}$.
  calculate $\mathbf{u}(s_{m+1}) = \mathbf{u}(s_m) + \frac{d\mathbf{u}}{ds}(s_{m+1} - s_m)$.
**end for**
**end procedure**
$\mathbf{u}(s = 1)$ is our sample from the CNF

---

