# OpenReview forum: "Characteristic Neural Ordinary Differential Equations"
_NeurIPS.cc/2022/Conference — NeurIPS 2022 Submitted_

### Official Review · Reviewer_E8Rb · 2022-06-13

**Rating:** 7
**Confidence:** 4
**Soundness:** 4 excellent
**Presentation:** 3 good
**Contribution:** 3 good

**Summary:**

The authors propose Characteristics NODE (C-NODE) which parametrize the method of characteristics ODE of Hamilton-Jacobi equations. The authors provide density estimates for C-NODE. The authors backs up their claim of improving performance with experiments on images and time series.

**Questions:**

- In algorithm session, only procedures for image processing tasks are discussed. What about the other ones?
- Also in algorithm session, what is your starting x(s=0)?
- Equations like burgers may produce shock or rarefactions during their solution. How do you deal with those?


**Limitations:**

MOC only applies to Hamilton-Jacobi equations (not just hyperbolic), but still, it does not apply to all equations and may fail to model some type of problems. This should not be an essential problem since it could be solved with Augmentation.

**Strengths And Weaknesses:**

Strengths:
- The idea of combining method of characteristics and NODE is very straightforward and new.
- The density estimates are theoretical supported and works well on CNF type problem in their experiments

Weakness:
- Section 3 is a little bit messy. It is easy to find out the idea behind, but it is a bit confusing on things like whether they want to solve the PDE numerically or the characteristics ODE numerically; how they will do interpolation after they solve with characteristics ODE, etc.
- Experiments on PDE modeling and time series prediction are synthetic data only. It might be converntional for PDE modeling, for time series prediction, this is not a very convincing experiment.
- For problems without a general PDE view, like image classification and time series prediction, the model is simply combination of equations (5) and (6), which is basically more augmentations on NODE, which may not be something new. Still, this provides an interesting PDE viewpoint that is very inspiring for CNF tasks.

---

> ### Author Response · Authors · 2022-08-02
> **Author Response**
>
> We thank the reviewer for the helpful review and the constructive comments.
> We answer the relevant questions below.
>
> [Question 1, Section 3 Clarity]
> We apologize for the confusion in this section.
> We have edited the manuscript in the revision to enhance clarity. The reviewer is correct that when solving the ODE corresponding to multiple characteristics we recover the full PDE, which is the computation that we perform in the method. Interpolation between solutions is done through conditioning the characteristic curves on different initial conditions.
>
> [Question 2, Additional Time Series Results]
> We agree with the reviewer that additional experiments would bolster the empirical evaluation.
> We have added these to the newest revision where we compare the proposed method to NODE on a subset of the MuJoCo data set. After 100 training epochs, C-NODEs achieve 10.14\% lower testing mean squared errors than NODEs on the prediction task on held out data.
>
> [Question 4, Missing Algorithms]
> We thank the reviewer for pointing this out, we have added additional algorithms in the appendix D of the revision.
>
> [Question 5, Characteristic Starting Value]
> The starting $x(s=0)$ is set to be 0 for all experiments.
>
> [Question 6, Shock waves and Rarefactions]
> The reviewer brings up a very interesting point.
> In practice, we did not enforce any particular structure to prevent characteristics from intersecting and inducing shockwaves.
> However, we believe that due to the high dimensionality of the ambient space that we consider, this is unlikely to happen.
> Rarefactions generally would not be an issue since the solution is always integrated to a point where a solution exists.
> We have modified the manuscript to include a discussion on this point.

---

> > ### Comment · Reviewer_E8Rb · 2022-08-03
> > **Response**
> >
> > Thanks for answering all my questions! Despite still lacking emperical support on performance on time series datasets, I'd recommend this paper.

---

> > > ### Author Response · Authors · 2022-08-03
> > > **Author Response**
> > >
> > > We thank the reviewer again for constructive comments, they really helped improving the paper!

---

### Official Review · Reviewer_1nEB · 2022-07-07

**Rating:** 6
**Confidence:** 4
**Soundness:** 4 excellent
**Presentation:** 3 good
**Contribution:** 3 good

**Summary:**

**Edit after Reviewer and AC discussion**

After a thorough discussion with the other reviewers and AC unfortunately I will lower my score to a 6. The main reason is that the most revealing experiment is in Table 1, which is a strong experiment. Revealing how all the baselines change with and without C-NODE. But image classification is not the best experiment for NODEs. Instead this strong comparison should be carried out on all experiments especially Normalizing Flows and Time-Series.


**Original**

This paper adapts the Neural Ordinary Differential Equation (NODE) framework to run along the characteristic curves of partial differential equations (PDES), the new model is called Characteristic Neural ODEs (C-NODEs).

This effectively is a different way to make more expressive NODE models, similar to Augmented Neural ODEs and Neural Controlled ODEs being techniques to improve expressivity and generalization of NODEs.

Proofs are provided to show this improves expressivity. C-NODEs are tested on classification and continuous normalizing flow tasks (density estimation) on MNIST, CIFAR-10 and SVHN. C-NODEs are also tested on time-series and PDE tasks. C-NODEs perform well on these tasks, outperforming NODEs as well as Augmented NODEs, 2nd Order NODEs and IL-NODEs.

**Questions:**

My only question is how best to view these C-NODEs, as they are solved with blackbox ODE solvers. Are they more like ODEs or PDEs? And if they are like ODEs is it possible for a NODE (or an ANODE/Latent NODE) to learn the same dynamics as C-NODE. Is it better to think of C-NODE as limiting the dynamics function so it has to be in the form of a characteristic curve? Similar to how convolutions share weights across linear layers to enforce translation symmetry.

**Limitations:**

The paper sufficiently explores its limitations. There is no broader impact statement, it isn't really required here, however, it is good to keep in mind that the applications of NODEs and C-NODEs can include some potentially unethical ones. This is because they can generally be applied to learning time-series and now partial differential equations. As said, this is not the case in the current form of the paper, the work is incremental and not applied in those directions.

**Strengths And Weaknesses:**

Strengths:

The paper is generally very good, with good contributions and methdology, specifically:
- The theoretical grounding of C-NODEs is good.
- The experiments on the whole are good, experimenting against expected baselines on standard benchmark tests.
- The paper is well written, with helpful examples for difficult/new concepts.

Weaknesses:

The largest weakness is that the paper generally feels a little incomplete, the work is brilliant but there are some experiments that could make it a brilliant paper. Specific examples are:

- There is this term $g(z)$ included in the dynamics, allowing the vector field to be conditioned on the input. It would be quite informative to see how much this term affects the answer. That is, can this term (a learnable function with the same structure) solve the classification problem on its own? This would make a convincing ablation if it can't.
- This term $g(z)$ also seems to give the initial condition for the C-NODE, it might be helpful to visualise the C-NODEs time evolution, if it stays roughly the same $g(z)$ is doing the majority of the work.
- Parameter efficiency is mentioned in the paper, but this doesn't seem to be explicitly tested, again this would make a good appendix study.
- Results are missing for testing Continuous Normalizing Flows (and discrete flows) on SVHN.
- The paper claims the adjoint method is better because it can be used with adaptive solvers due to memory efficiency. The Euler method is used with direct backprop. It would be helpful to show that Dopri5 fails when using direct backprop because memory runs out.
- It is noted that coupling C-NODEs with ANODEs can make training more stable, apart from in the classification experiments it doesn't look like this has been tested.

Aside from this there are some minor typos or tiny mistakes that do not affect the quality of the paper but for the authors knowledge:

- Line 26-27: It might help to clarify the adjoint method is only memory efficient in integration time, and not size of the dynamics function for example.
- Between lines 52 and 53: I would change "would take too much memory" to "would require too much memory".
- Line 85: "then its log likelihood from Chen et al.:" should be changed to "then its log likelihood from Chen et al. is given by:"
- Line 107: Implies $s \in [0, 1]$ when it seems that $s \in[0, T]$
- Line 200: "NFE is a indicator" should be "NFE is an indicator"
- Line 230: "Differential equations are solved using the adjoint method and a Runge-Kutta of order 5 of the Dormand-Prince-Shampine solver." Should be "Differential equations are solved with a Runge-Kutta of order 5 of the Dormand-Prince-Shampine solver and trained with the adjoint method".
- Line 236: "using a lower NFEs" should be "using a lower NFE".
- Line 265: "and can combine with" should be "and can be combined with"

---

> ### Author Response · Authors · 2022-08-02
> **Author Response**
>
> We thank the reviewer for the helpful comments and insightful review of our manuscript.
> We respond to the questions below.
>
> [Question 1, structure of $g(z)$]
> We apologize for the confusion regarding $g(z)$ in the empirical evaluation.
> For all of the classification experiments, we set $g(z) = z$ which makes the input into C-NODE the original image.
> By leaving the initial conditions constant, we can focus exclusively on the performance of C-NODE itself without influence from the network that is transforming the input.
> We believe that by making $g(z) \neq z$ the performance of the method would improve, but, as the reviewer rightly points out, this would obfuscate the contributions of C-NODE versus the network $g$.
> We have updated the manuscript to emphasize this in the main text.
> The only experiment where the initial condition is calculated with $g(z) \neq z$ is for the PDE experiment due to the necessity of modeling the boundary condition.
>
> [Question 2, parameter efficiency]
>
> We agree with the reviewer that this would be a good ablation. We added an ablation study on the parameters used in C-NODE and NODE on the image classification task with the CIFAR-10 dataset. C-NODE consistently achieves better results than NODE across the different number of parameters used along the whole training process.
>
> [Question 3, Results missing on SVHN]
> While we compare the results of C-NODE on generative modeling for SVHN to the continuous normalizing flow based on NODE, we did not compare to the discrete normalizing flows methods because we could not find prior work that compared to SVHN.
> Given the time constraints, we are unable to perform the experiments ourselves to report the values, but we will update the final manuscript to include these.
>
> [Question 4, DOPRI5 Memory]
> We agree that the memory issue for direct backpropogation should be better described.
> Due to the adaptive step size of DOPRI5, the number of steps for any given time interval can be quite large.
> In practice, we witnessed memory errors when trying to backpropagate through the individual solver steps.
> In the table of the classification results, we mentioned the number of function evaluations.
> Following Table 1 given in [1], the number of function evaluations roughly corresponds to the number of layers in a feed forward network and describes the order of the memory needed.
> Since most of the number of function evaluations is generally always greater than 50 and often over 100, one can see that this incurs a large memory footprint.
> We have updated the manuscript to emphasize this issue.
>
> [1] Chen, Ricky TQ, et al. "Neural ordinary differential equations." Advances in neural information processing systems 31 (2018).
>
> [Question 5, Stability with ANODE]
> We agree with the reviewer that considering experiments with augmentation would be beneficial.
> The other set of large scale experiments that we conduct are the density estimation experiments.
> However, since ANODE requires lifting the latent variable to a higher dimension, the direct application of the change of variables formula to describe the transport of the base density to the target density is not straightforward since it requires defining the notion of the augmenting dimension.
> For these reasons we did not include a comparison of the stability for ANODE in the density estimation experiments.
> However, this does make an interesting direction for future work.
>
> [Question 6, Interpretation of C-NODEs]
> We view C-NODE as a solver for PDEs, as the reviewer mentions, with a particular limited form on the PDE structure.
> This is because the decomposition of the terms in the integrand leads to solving a PDE over different sets of curves, namely the characteristics.
> In that sense, the dynamics are limited because they must share the same $J_x u$ for each point, similar to convolution layers in computer vision tasks.
> On the other hand, since the PDE over a particular characteristic is reduced to an ODE, an ODE interpretation can be useful.
> However, taking only the ODE viewpoint disregards the interaction between the different ODE solutions which is a key point of using the method of characteristics.

---

> > ### Comment · Reviewer_1nEB · 2022-08-03
> > **Thank you for your answers**
> >
> > Dear authors,
> >
> > You have sufficiently answered my questions. And I will keep my recommendation at accept.
> >
> > If accepted for the camera ready version, please include discrete normalizing flows results on SVHN. Also the parameter ablation in the appendix is good, again it could be improved by running it against the other baselines of the paper (ANODE, IL-NODE).

---

> > > ### Author Response · Authors · 2022-08-03
> > > **Author Response**
> > >
> > > Dear Reviewer,
> > >
> > > We appreciate your highly constructive suggestions and enlightening comments.
> > >
> > > Indeed, adding the discrete normalizing flows results on SVHN and parameter ablation on more NODE type algorithms will further improve the paper. If accepted, we will make sure to add those in the final manuscript.

---

### Official Review · Reviewer_EacU · 2022-07-20

**Rating:** 4
**Confidence:** 3
**Soundness:** 3 good
**Presentation:** 2 fair
**Contribution:** 2 fair

**Summary:**

The authors investigate the role of the method of characteristics (MoC) on the modeling of the latent variables of NODEs, and propose a new algorithm named C-NODEs. C-NODEs parameterize the latent variables of NODEs as the solution of a family of first-order quasi-linear PDEs along the characteristic curves. The authors prove C-NODEs can learn intersecting trajectories and also are universal approximators of homeomorphisms. Experiments empirically show C-NODEs improve the learning accuracy and efficiency on multiple tasks.

**Questions:**


1. This paper uses MoC to parameterize or augment the latent variables of NODEs. However, this parameterization restrict the latent varialbes to the solution of first-order quasi-linear hyperbolic PDEs. Does this hypothesis make sense?

2. Some other papers also propose ways to augment NODEs by parameterize the latent variables, such as ANODE, Second-order NODEs (SONODE). Are C-NODEs better than these methods?

3. In Equation (7) (line 129), the functions $J_x u$ and $dx/ds$ are modeled by neural networks. Using $\text{NN}_1$ and $\text{NN}_2$ to represent neural networks, is it equivalent to $u(x(T)) = u(x(0)) + \int_0^T \text{NN}_1(x,u;\Theta_2) \text{NN}_2(x,u;\Theta_2)ds$? If so, the structure of Equation (4) will not be preserved.

4. Near the line 111, $\frac{d}{ds}u(x(s),t(s)) = \frac{\partial u}{\partial t} + u \frac{\partial u}{\partial x} = 0$. If so, by integrating over $s$, we have $u(x(T),t(T);x_0,t_0) \coloneqq \int_0^T \frac{d}{ds} u(x(s),t(s))ds = \int_0^T 0 ds = 0$ ?

typo: line 3 - ``a latent variables"

**Limitations:**

Yes

**Strengths And Weaknesses:**

The expressiveness of vanilla NODEs is indeed a problem due to the restriction of ODEs. The paper try to involve the family of PDEs to augment NODEs. The paper is easy to follow. The strength of this paper is the rich experiments and analysis of the properties of C-NODEs. However, the presented work raises several questions. The main weakness is the clarity and insufficient baselines. See questions below.

---

> ### Author Response · Authors · 2022-08-02
> **Author Response**
>
> We thank the reviewer for the time and effort in providing feedback to strengthen the manuscript.
> We provide individual responses to questions below.
>
> [Question 1, MoC Applicability]
> The reviewer makes a good point that the MoC does not apply to all types of PDEs, and we agree with the reviewer that the motivation behind this family of PDEs should be better described.
> The most general qualitative description of this family of equations is a transport equation -- which roughly describes the propagation of certain quantities through time.
> Such equations are appropriate for deep learning tasks due to their ability to transport data into different regions of the state space.
> For example, in a classification task, we consider the problem of transporting high-dimensional data points that can not be linearly separated to spaces where they can be linearly separated.
> Similarly in a generative modeling task, we transport a Gaussian distribution to data distribution.
> In that sense, we believe this family of equations is sufficient for relevant tasks in machine learning.
> However, as the reviewer points out, more general types of PDEs cannot be represented by C-NODE, which is a limitation of the method.
> We have updated the manuscript to include this description and limitation.
>
> [Question 2, Augmenting NODEs]
> We note that C-NODE can be applied to the augmented forms, as we demonstrated in Table 1 when modifying ANODE for use with C-NODE. We also add additional experimental results comparing C-NODE, ANODE, and NODE on time series prediction tasks, as shown below
>
> | Time   | [0,1]  | [1,2]  | [2,3]  | [3,4]  | [4,5]  | [5,6]  |
> |--------|--------|--------|--------|--------|--------|--------|
> | NODE   | 0.0322 | 0.1764 | 0.4681 | 0.8093 | 1.1911 | 1.6202 |
> | ANODE  | 0.0428 | 0.0629 | 0.1248 | 0.2778 | 0.5360 | 0.9252 |
> | C-NODE | 0.0270 | 0.0365 | 0.0582 | 0.1474 | 0.3300 | 0.6054 |
>
> | Noise  | 0      | 1      | 2      | 3      | 4      | 5      |
> |--------|--------|--------|--------|--------|--------|--------|
> | NODE   | 0.0326 | 0.1784 | 0.7886 | 1.9685 | 3.7530 | 6.1553 |
> | ANODE  | 0.04   | 0.1984 | 0.6035 | 1.0574 | 1.4850 | 2.0593 |
> | C-NODE | 0.0267 | 0.1011 | 0.3294 | 0.7148 | 1.2856 | 2.0834 |
>
> We test C-NODEs, ANODEs, and NODEs on a synthetic time series prediction problem.
> We define a function by $u(x,t)=\frac{2x \exp(t)}{2\exp(t)+1}$, and we sample $\tilde{u} = u(x,t) + 0.1\epsilon_t$, where $\epsilon_t \sim \mathcal{N}(0, 1)$ over $x\in [1,2]$, $t\in[0,1]$ to generate the training dataset.
> We test the performance on $t \in [n,n+1]$ with $n\in\{0,1,\ldots,5\}$.
> We also test C-NODEs, NODEs, and ANODEs on time series prediction with different levels of noise. Specifically, using the same function as above, we form training and testing dataset with $\epsilon_t\sim \mathcal{N}(0,m)$, $m\in\{0,1,\ldots,5\}$. We test the performance on the time period $t\in[0,1]$. In both cases, we report the testing mean squared errors.
>
> Since SONODE is an alternative interpretation of ANODE, the same principle applies.
> One difference between C-NODE and ANODE/SONODE is that both ANODE/SONODE augment the dimension, and directly applying the change of variables formula for continuous normalizing flows is not straightforward due to the augmenting dimensions.
> We have modified the manuscript to emphasize these points.
>
> [Question 3, Two Neural Networks]
> We agree with the reviewer that there is a level of unidentifiability within the proposed framework since both components are represented as neural networks.
> We note that the network architecture is factored in such that one component describes the Jacobian and the other describes the characteristic.
> This factorization enforces that the Jacobian is shared for all data points but the characteristics are modified for different data points.
>
> [Question 4, Burgers Equation]
> The reviewer is correct that, for the case of the Burgers equation, the value of the solution does not change along the characteristic and is given by $u_0 + \int_0^T 0 ds$.
> However, this need not be the case since in general, the right hand side is not equal to zero.
> Specifically, as shown in equation (4), (5), and (6), the right hand side of the PDE is $c(x_1,...,x_k,u)$, and integrating along $s$ would result in $$\int_0^T\frac{d}{ds}u(x_1,...,x_k,u)ds=\int_0^Tc(x_1,...,x_k,u)ds.$$

---

> ### Author Response · Authors · 2022-08-05
> **A Gentle Reminder**
>
> Dear Reviewer EacU
>
> We would like to thank you again for the time you dedicated to reviewing our paper and for your valuable comments. We believe that we have addressed your concerns. Since the end of the discussion period is getting close and we have not heard back from you yet, we would appreciate it if you kindly let us know of any other concerns you may have, and if we can be of any further assistance in clarifying any other issues.
>
> Thanks a lot again, and with sincerest best wishes,
>
> Authors

---

> ### Author Response · Authors · 2022-08-08
> **A Second Gentle Reminder**
>
> Dear Reviewer EacU
>
> We apologize for any inconvenience that our message may cause in advance.
>
> Again, we would like to thank you for the time you dedicated to reviewing our paper and for your valuable comments. We believe that we have addressed your concerns.
>
> Since the end of the discussion period is close and we have not heard back from you yet, we would appreciate it if you kindly let us know of any concerns you may have, and if we can be of any further assistance in clarifying any other issues.
>
> We humbly remain at your disposal.
>
> Thanks a lot again, and with best wishes
>
> Authors

---

### Meta-Review · Area_Chair_2sks · 2022-08-26

**Recommendation:** Reject
**Confidence:** Less certain

**Metareview:**

This paper proposed to model the evolution of the latent variables to the characteristic curves instead of the original ODEs. Authors proved the new method C-NODE is more expressive than the original NODE. Experiments are conducted on image classification tasks to demonstrate its effectiveness. It will be good explorations to leverage the differential equation theory to improve the NODE algorithms. The insights from the journey will help innovate breakthrough directions in operator learning. During the discussion phase, reviewers had rounds of debates about whether the method is demonstrated effective on standard tasks for NODEs.  Although it is a high bar for exploration style work to achieve SOTA results,  we expect some insights from the investigation. For example, why the former NODE is not expressive enough in certain tasks, what might be the factors in the real tasks influencing the expressiveness, why the image classification task need extra expressiveness, etc. Simulations can also be involved to show these insights in extreme cases.

**Award:**

No

---

### Decision · Program_Chairs · 2022-09-14

Reject